# Are Large Language Models Good Statisticians?

**Yizhang Zhu**[1], **Shiyin Du**[1], **Boyan Li**[1], **Yuyu Luo**[1,2]*, **Nan Tang**[1,2]*

[1]The Hong Kong University of Science and Technology (Guangzhou), Guangzhou, China
[2]The Hong Kong University of Science and Technology, Hong Kong SAR, China
{yzhu305, sdu164}@connect.hkust-gz.edu.cn
{boyanli, yuyuluo, nantang}@hkust-gz.edu.cn

## Abstract

Large Language Models (LLMs) have demonstrated impressive capabilities across a range of scientific tasks including mathematics, physics, and chemistry. Despite their successes, the effectiveness of LLMs in handling complex statistical tasks remains systematically under-explored. To bridge this gap, we introduce `StatQA`, *a new benchmark designed for statistical analysis tasks*. `StatQA` comprises 11,623 examples tailored to evaluate LLMs' proficiency in specialized statistical tasks and their applicability assessment capabilities, particularly for hypothesis testing methods. We systematically experiment with representative LLMs using various prompting strategies and show that even state-of-the-art models such as *GPT-4o* achieve a best performance of only 64.83%, indicating significant room for improvement. Notably, while open-source LLMs (*e.g., LLaMA-3*) show limited capability, those fine-tuned ones exhibit marked improvements, outperforming all in-context learning-based methods (*e.g., GPT-4o*). Moreover, our comparative human experiments highlight a striking contrast in error types between LLMs and humans: LLMs primarily make applicability errors, whereas humans mostly make statistical task confusion errors. This divergence highlights distinct areas of proficiency and deficiency, suggesting that combining LLM and human expertise could lead to complementary strengths, inviting further investigation into their collaborative potential. Our source code and data are available at `https://statqa.github.io/`.

## 1   Introduction

Statistical analysis can capture data patterns and convert them into usable evidence, which is crucial to data science and machine learning applications [1–8]. As shown in Figure 1, a typical statistical analysis task involves, given a table $D$ and a statistical question $Q$, a qualified statistician should be proficient in selecting relevant columns $\mathbb{C}$, choosing the appropriate statistical methods $\mathbb{M}$, and computing the results based on $\mathbb{M}$ using $\mathbb{C}$. Formally, the statistical analysis task can be divided into two stages: (1) identifying appropriate statistical methods and parameters (*e.g.,* relevant columns in the table), and (2) computing the statistical results and deriving the conclusion. The first stage requires statistical expertise to assess the applicability of methods, considering factors such as data type, distribution characteristics, and sample size, which is the core of the statistical task. In contrast, the second stage (*i.e.,* computation) can be easily aided by external tools [9] such as WolframAlpha [10].

Inspired by the extensive application of Large Language Models (LLMs) [11–13], we pose a critical question: *Do LLMs truly understand such "statistical literacy"?* Specifically, can LLMs competently select relevant data, recognize prerequisites, and discern appropriate usage scenarios to assess the effectiveness of statistical methods?

---

*Yuyu Luo and Nan Tang are the corresponding authors.

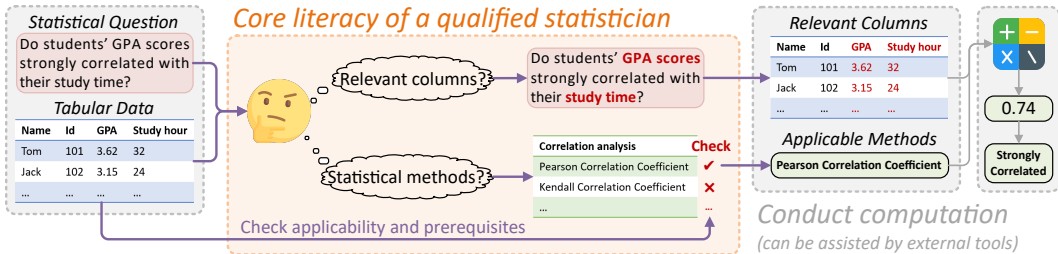

Figure 1: An Example of Statistical Analysis Task

Currently, research on the mathematical reasoning abilities of LLMs has primarily focused on the accuracy of computational processes and results, involving conventional statistical methods [14–17]. However, studies examining datasets requiring specialized statistical testing methods, particularly those assessing the applicability of these methods, are minimal. This gap in the literature motivated us to explore and address this area. Our study aims to answer the following critical questions.

- **Q1:** *How can we evaluate LLMs' performance in more complex and specialized statistical testing tasks?* Constructing an appropriate benchmark is crucial for accurate performance evaluation. Given the challenges posed by the lack of datasets and scarcity of examples in this specialized field, how can we efficiently develop such a benchmark?
- **Q2:** *How capable are current LLMs in this field, and how can we improve their performance?* This requires systematic experiments to evaluate the capabilities of current LLMs, exploring the impact of different models, prompting strategies, and fine-tuning methods on their performance.
- **Q3:** *How do humans perform compared to LLMs, and what are the differences in their performance?* This involves a comparative study between humans and LLMs, aiming to analyze their respective strengths and weaknesses and explore potential complementary between human expertise and LLM capabilities.

**Contributions.**  Our contributions are summarized as follows:

- **StatQA.** We propose `StatQA`, a new benchmark for statistical analysis tasks, particularly focusing on the applicability assessment of statistical methods. We introduce an automated pipeline to construct `StatQA` by synthesizing statistical tasks and their corresponding answers, which also provides insights for dataset construction in other specialized domains with scarce examples.
- **Systematic Evaluation.** We conduct extensive evaluations on widely used LLMs to establish benchmarks for statistical tasks. We also explore several strategies, including domain-specific prompts and fine-tuning, to better harness the capabilities of LLMs for these tasks.
- **Comparative Study between Humans and LLMs.** We organize group-based human experiments and comparatively analyze differences between humans and LLMs in performance and errors. Our findings highlight humans' and LLMs' distinct strengths and weaknesses and reveal their potential complementarity.
- **New Empirical Findings and Research Opportunities.** Based on the experiments and analysis above, we summarize six key findings and discuss research opportunities in this field.

## 2   StatQA

In this section, we will first discuss the design goal of the benchmark for statistical analysis tasks (Section 2.1). We will then describe the characteristics of `StatQA` (Section 2.2). Finally, we will elaborate on how to develop `StatQA` with low human cost while ensuring high quality (Section 2.3).

### 2.1   Design Goals and Tasks Scope

**Goals.**  We aim to develop a specialized dataset to address the current research gap. Our goals include: (G1) *Good Coverage of Statistical Tasks and Difficulties*: The benchmark should encompass representative and commonly used statistical analysis tasks and methods. It should be designed to be sufficiently discriminative, effectively distinguishing between the strengths and weaknesses

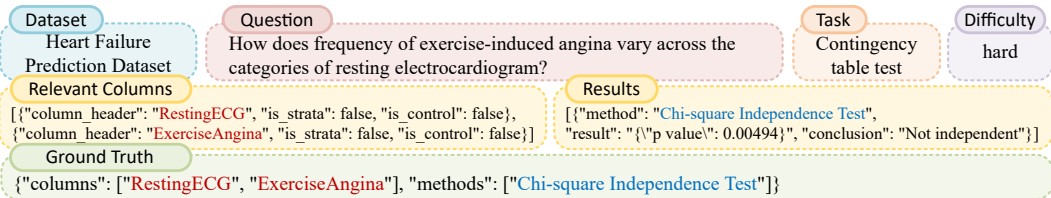

Figure 2: An Example in StatQA. (1) Dataset ($D$) is the datasets for statistical tasks; (2) Question ($Q^*(\mathbb{C})$) is a statistical question, refined by GPT-3.5-Turbo; (3) Task indicates the statistical task category; (4) Difficulty includes easy and hard levels. (5) Relevant Columns ($\mathbb{C}$) are data columns relevant to the statistical question; (6) Results ($\mathbb{R}$) include applicable statistical methods ($\mathbb{M}$), the computational results, and preliminary conclusions; (7) Ground Truth includes relevant columns ($\mathbb{C}$) and all applicable methods ($\mathbb{M}$) for this statistical task;

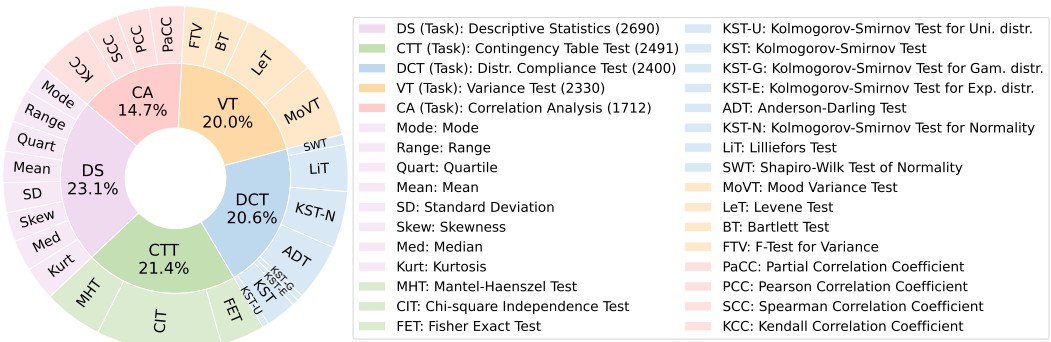

Figure 3: The Proportion of Statistical Tasks in StatQA. The inner ring represents the five types of statistical tasks mentioned, while the outer ring corresponds to the methods associated with each task.

Table 1: Statistics of StatQA

| Item | Tabular Data | | Question Length (Chars) | | | Difficulty | | #-Examples | |
|---|---|---|---|---|---|---|---|---|---|
| | Avg #-Rows | Avg #-Cols | Max | Min | Avg | Easy | Hard | StatQA | mini-StatQA |
| **Stats** | 6,228 | 14 | 346 | 21 | 113 | 7,401 | 4,222 | 11,623 | 1,163 |

of different LLMs; (G2) *Support for Evaluating Statistical Literacy*: The dataset should facilitate our core experiments, assessing whether LLMs can evaluate the applicability of statistical methods, select suitable methods, and identify relevant data columns; (G3) *Low Human-Cost*: Automating the dataset construction process to ensure sufficient scale and improved efficiency while maintaining high data quality; and (G4) *Extensibility*: The benchmark should be designed to accommodate future expansions and enhancements.

**The Scope of Statistical Analysis Tasks.** Statistical tasks can be broadly divided into descriptive statistics and inferential statistics, with inferential statistics primarily including regression analysis and hypothesis testing [18]. Descriptive statistics and hypothesis testing represent two prevalent methods frequently employed in statistical analysis. The latter is often considered the most misunderstood in quantitative analysis due to its complex interdependencies between procedural components [19]. Therefore, together with Descriptive Statistics (DS), we select four representative categories of statistical tasks in hypothesis testing to be covered in StatQA, along with commonly used methods: Correlation Analysis (CA); Contingency Table Test (CTT); Distribution Compliance Test (DCT); Variance Test (VT). The involved statistical tasks in each category are shown in Table 3 (Section B.2).

## 2.2 StatQA Characteristics

Figure 2 shows an example in StatQA, which includes the dataset (tabular data), statistical question, task category, difficulty level, relevant column information, preliminary results, and ground truth. More examples are presented in Figure 7 in the appendix.

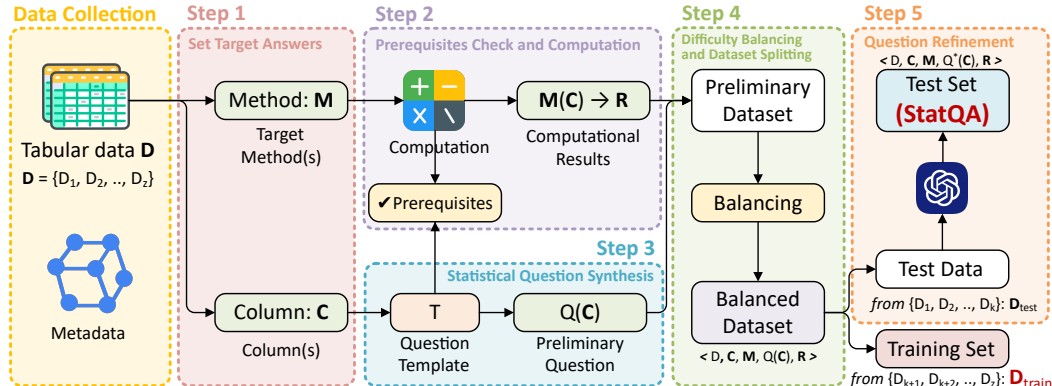

Figure 4: The Pipeline for Synthesizing StatQA

Figure 3 and Table 1 show the proportion of different statistical task categories and the statistics of our `StatQA`, respectively. The `StatQA` benchmark contains 11,623 examples. To reduce testing costs and facilitate users with limited computational resources, we use the *stratified sampling* strategy [20] to obtain the mini-`StatQA` (1,163 examples), ensuring mini-`StatQA` resembles the complete benchmark in terms of task and difficulty distribution. We also use mini-`StatQA` in subsequent experiments.

## 2.3 StatQA Construction

**Key Ideas for Developing StatQA.** In conventional dataset construction, researchers collect a suitable dataset $D$, formulate a question $Q$, and manually annotate answers $A$. While this method ensures high data quality, it is time-consuming, costly, and limits extensibility, especially in specialized domains with scarce examples. To alleviate these limitations, our **key idea** is to reverse this process by synthesizing the question $Q$ based on target answers $A$. We start with target answers $A$ derived from the tabular data $D$ and generate corresponding statistical questions $Q$. This approach ensures precise alignment between questions and answers, enabling more efficient dataset construction.

To implement this, we design an efficient pipeline for constructing `StatQA`, as shown in Figure 4. Unlike traditional methods, we set target answers $A$ based on tabular data $D$ and then synthesize statistical questions $Q$ in reverse. To ensure alignment between $Q$ and $A$, we incorporate automated prerequisite checks. To support the evaluation of statistical literacy, the target answers $A$ include relevant columns $\mathbb{C}$ and applicable statistical methods $\mathbb{M}$, enabling the derivation of computational results $\mathbb{R}$. Therefore, our pipeline can synthesize numerous examples of $(D, \mathbb{C}, \mathbb{M}, Q, \mathbb{R})$ along with other supplementary information. Next, we will go through our pipeline step by step.

**Tabular Data and Metadata Collection.** We collect 78 tables from real-world applications, covering various domains including education, medicine, economy, etc., as shown in Section B.5. These tables are carefully gathered by post-graduate students in statistics from Kaggle [21] and Rdatasets [22]. The metadata includes descriptive information about the tabular data and details of each data column. Specifically, it encompasses the column header, data type, normality, and description, which are crucial for prerequisite checks in statistical analysis. Descriptive information in metadata can be obtained from their sources, while the data types and normality of each column can be calculated and derived from the tabular data. For tables lacking metadata, we perform manual annotations. For tables with existing metadata, we conduct manual validation to ensure accuracy.

**Step 1: Set Target Answers (Select Target Methods and Columns).** As shown in Figure 4, our pipeline can reversely synthesize the statistical question $Q$ based on statistical methods and involved data columns. The first step is to set the target methods $\mathbb{M}$. Considering the parameter volume of the target methods, we select suitable columns from tabular data to obtain a set of data columns $\mathbb{C}$.

**Step 2: Prerequisites Check and Computation.** Since statistical methods should be used under appropriate conditions, we perform a series of checks to ensure all prerequisites are met. This includes verifying features such as sample size, data type, and normality. Suppose data columns $\mathbb{C} = \{C_1, ..., C_m\}$ fit with the prerequisites of methods $\mathbb{M} = \{M_1, ..., M_n\}$. The data columns $\mathbb{C}$

can then be used as parameters in the target methods $\mathbb{M}$ to compute the corresponding computational results and preliminary conclusions, noted as $\mathbb{R} = \{M_1(\mathbb{C}), ..., M_n(\mathbb{C})\}$.

**Step 3: Statistical Question Synthesis.** To ensure the quality of the questions, we use *hand-crafted* question templates to synthesize preliminary statistical questions. The templates are determined by the target method, listing common question expressions with placeholders where relevant data columns are involved, as shown in Section B.3. By selecting a template $T$ and substituting in previously chosen data columns $\mathbb{C}$, we can obtain the preliminary statistical question $Q(\mathbb{C})$.

**Step 4: Difficulty Balancing and Dataset Splitting.** For a statistical question, if the set of applicable methods constitutes a proper subset of the candidate methods (*i.e.,* all methods for comparable scenarios), it indicates that certain prerequisites for some methods are not satisfied. In such cases, we label the question as "hard", reflecting the increased challenge of accurately assessing applicability to eliminate inapplicable methods. Otherwise, the question is labeled as "easy". Based on these labels, we expand the underrepresented categories and sample the abundant ones to obtain balanced synthesized examples in terms of task difficulty. Next, we split these synthesized examples to ensure no table overlaps in training and test sets: data synthesized from source tables No. 1 to 36 will be used for subsequent tests (*i.e.,* StatQA), while tables No. 37 to 78 will be used for training (*i.e.,* training set $\mathbb{D}_{\text{train}}$). The $\mathbb{D}_{\text{train}}$ is used to fine-tuning LLMs.

**Step 5: Statistical Question Refinement.** Inspired by LLMs' capabilities in text comprehension and processing, we use GPT-3.5-Turbo to refine the phrasing and expression of statistical questions for the test data. We provide GPT-3.5-Turbo with original questions and their descriptive information, instructing GPT-3.5-Turbo to paraphrase and refine the question sentences without changing the meaning, aiming for more coherent and diverse expressions, which is noted as $Q^*(\mathbb{C})$.

**Discussion of Quality Control.** We devise several strategies to ensure the high quality of our StatQA. (1) *Quality of Question Templates*: The quality of question templates is a focal point in our quality control. We categorize templates based on statistical tasks and applicable scenarios of methods. For each category, we meticulously prepare 10 to 20 templates for random selection to enhance diversity. To ensure the templates are representative, we recruited two post-graduate students in statistics to design and review them. (2) *Question Refinement*: The objectives of refinement are to correct potential grammar mistakes, improve semantical coherence, and increase the diversity of expressions for the questions generated by the templates. In practice, GPT-3.5-Turbo demonstrates a satisfactory level of English proficiency in achieving these goals and offers better cost-effectiveness; therefore, we employ GPT-3.5-Turbo as the refiner. Note that question refinement is exclusively performed on StatQA to increase diversity and ensure differences from the training set. The average BLEU [23], BERTScore [24] calculated between original questions and refined versions in StatQA is 0.126, 0.920 respectively. The low BLEU reflects notable vocabulary differences, while the high BERTScore suggests semantic consistency, collectively indicating adequate rephrasing that preserves the original meaning. (3) *Expert Reviews*: Last but most importantly, we conduct manual reviews by two post-graduate students in statistics to carefully check all examples in StatQA.

## 3 Experiments

### 3.1 Setup

**Experimental Protocols.** We design experiments for LLMs similar to human statisticians' mindset, as presented in Figure 1, to evaluate the abilities of LLMs in statistical tasks. Because of the limitation of input tokens, we provide LLMs column information instead of the whole table, including the column headers, number of rows, data type, and normality. In the experiment, the LLMs need to pick headers of relevant data columns, assess the methods' applicability, select all statistical methods that fit the usage scenario and prerequisites as statisticians, and then respond in a specific format. Since LLM responses might be invalid or include irrelevant content, cleaning, and extraction are necessary, so the extracted answers should be compared to the ground truth for evaluation. In the human experiments, we use the same protocol for consistency and develop a testing platform to facilitate participant selection. *More details of experimental setups, including hyperparameters,*

*prompts, procedures, and GUI of the testing platform used in human experiments, are provided in Section C.*

**Metrics.** Accuracy of relevant data columns and applicable methods selections, noted as $Acc(\mathbb{C}, \mathbb{M})$, is used as our metrics to evaluate if LLMs or participants truly understand the question and the applicability of statistical methods. $Acc(\mathbb{C}, \mathbb{M})$ refers to the proportion of methods and column selections fully aligned with the ground truth without any omissions or incorrect selections:

$$Acc(\mathbb{C}, \mathbb{M}) = \frac{\sum_{i=1}^{N} \mathbb{1}(\hat{\mathbb{C}}_i = \mathbb{C}_i, \hat{\mathbb{M}}_i = \mathbb{M}_i)}{N}, \tag{1}$$

where $N$ is the total number of test examples.

### 3.1.1 Evaluation for LLMs

**Non-fine-tuned LLMs.** For open-source LLMs, we select and conduct experiments on LLaMA-2 models (Llama-2-7b-chat-hf, Llama-2-13b-chat-hf) [25] and LLaMA-3 models (Meta-Llama-3-8B, Meta-Llama-3-8B-Instruct) [26]. For proprietary LLMs, we select representative and widely-used ChatGPT (GPT-3.5-turbo) [27], GPT-4 [28] and newly released GPT-4o [29].

**Prompting Strategies.** Few-shot learning and Chain-of-Thought (CoT) [30] will be used as prompting strategies. In few-shot learning, we prepare one example of each task category for the LLMs to learn from. To prevent the leakage of actual task categories, we randomly select examples in the few-shot. Furthermore, we introduce a new strategy to include domain knowledge (DK) of statistical methods' applicability and prerequisites in the prompt.

**Fine-tuned LLMs.** We fine-tune three models: LLaMA-2 (LLaMA-2-7b-chat-hf) and LLaMA-3 (Meta-Llama-3-8B, Meta-Llama-3-8B-Instruct). For all fine-tuning, we use the LoRA [31] method, which is a Parameter-Efficient Fine-Tuning technique that adjusts only a small number of parameters. This method achieves an effect close to full-parameter fine-tuning on downstream tasks while reducing computation and storage costs. Note that `StatQA` is exclusively reserved for testing and evaluation. We use the training set, also generated by our dataset construction pipeline (see **Step 4** in Section 2.3), to fine-tune the LLMs.

### 3.1.2 Human Experiments

**Participants.** Human experiments are conducted for a comparative study. We recruited 6 post-graduate students and grouped them based on their disciplinary backgrounds: Non-Statistics Background Group (Non-Stats, three STEM post-graduate students not in statistics major) and Statistics Background Group (Stats, three post-graduate students in statistics major).

**Protocols.** We use stratified sampling to extract 10% of the mini-`StatQA` for human experiments (117 examples). To ensure participants understand the task and are familiar with operations on our testing platform, they are required to watch a tutorial video before starting. Each participant needs to use 2 answering modes respectively during the experiments: (1) Closed-book: participants must answer independently; (2) Open-book: similar to introducing domain knowledge for LLMs, participants are provided with supplemental information. More details can be found in Section C.2.

### 3.2 Experimental Results and Analysis

Different models display significant performance discrepancies on our benchmark, indicating sufficient discriminative ability. Table 2 presents the experimental results. Figure 5 shows the radar chart depicting performances across five task categories achieved by leading results within each model.

**Overall Performance of LLMs.** The LLaMA-2 models demonstrate weak performance, suggesting their inability. The newer LLaMA-3 shows remarkable improvement over the previous LLaMA-2, with the LLaMA-3-8b achieving an $Acc(\mathbb{C}, \mathbb{M})$ up to 36.1%, far surpassing all tested LLaMA-2 models, close to the performance of 0-shot GPT-3.5-Turbo (37.4%). However, it is still noticeably weaker than GPT-4 and GPT-4o. By fine-tuning, LLaMA-2-7b, LLaMA-3-8b, and LLaMA-3-8b-Instruct models achieve marked enhancement, with the fine-tuned LLaMA-3-8b model showing the best overall performance, but considerable room still remains for further improvement.

Table 2: Experimental Results of $Acc(\mathbb{C}, \mathbb{M})(\%)$ on mini-StatQA. The `1st`, `2nd`, `3rd` place results in all experiments are highlighted in red, blue, and green respectively. The **bold** results are the best in each section. The underlined results are the leading ones of each subgroup in overall $Acc(\mathbb{C}, \mathbb{M})$, whose performances in sub-tasks are shown in Figure 5. CA: Correlation Analysis; CTT: Contingency Table Test; DCT: Distribution Compliance Test; VT: Variance Test; DS: Descriptive Statistics.

| Model | Strategy | Overall | CA | CTT | DCT | VT | DS |
|---|---|---|---|---|---|---|---|
| *Open-source LLMs: LLaMA-2/3* | | | | | | | |
| LLaMA-2 7B | 0-shot | 8.08 | 1.79 | 1.17 | 2.12 | 6.97 | 25.48 |
| | 1-shot | 14.96 | 0.60 | 6.25 | 5.93 | 19.26 | 37.07 |
| | 0-shot-CoT | 6.36 | 1.19 | 0.78 | 2.12 | 5.74 | 19.69 |
| | 1-shot-CoT | 14.45 | 1.79 | 4.30 | 8.48 | 19.67 | 33.21 |
| | 1-shot+DK | 16.08 | 0.60 | 7.42 | 9.32 | 18.44 | 38.61 |
| LLaMA-2 13B | 0-shot | 9.29 | 1.79 | 0.39 | 8.48 | 3.28 | 29.34 |
| | 1-shot | 17.97 | 9.52 | 5.47 | 9.32 | 2.05 | 58.69 |
| | 0-shot-CoT | 9.03 | 2.38 | 0.00 | 9.32 | 2.87 | 27.80 |
| | 1-shot-CoT | 17.63 | 6.55 | 9.38 | 9.75 | 0.41 | 56.37 |
| | 1-shot+DK | 20.29 | 8.33 | 7.03 | 16.53 | 11.48 | 52.90 |
| LLaMA-3 8B | 0-shot | 23.56 | 1.19 | 0.00 | 16.53 | 16.39 | 74.52 |
| | 1-shot | 31.90 | 17.86 | 8.20 | 18.64 | 25.41 | 82.63 |
| | 0-shot-CoT | 22.01 | 1.19 | 0.39 | 15.68 | 13.93 | 70.27 |
| | 1-shot-CoT | 32.24 | 14.29 | 5.86 | 19.92 | **29.10** | **84.17** |
| | 1-shot+DK | **36.11** | **26.79** | 20.31 | **29.24** | 15.98 | 83.01 |
| LLaMA-3 8B Instruct | 0-shot | 13.67 | 10.12 | 13.28 | 5.09 | 1.23 | 35.91 |
| | 1-shot | 28.20 | **26.79** | 12.11 | 13.56 | 9.84 | 75.68 |
| | 0-shot-CoT | 11.61 | 10.71 | 14.84 | 6.78 | 0.00 | 24.32 |
| | 1-shot-CoT | 28.29 | 26.19 | 16.80 | 16.10 | 9.84 | 69.50 |
| | 1-shot+DK | 27.77 | 19.64 | **22.27** | 20.34 | 8.20 | 63.71 |
| *Proprietary LLMs: GPT-3.5-Turbo, GPT-4 and GPT-4o* | | | | | | | |
| GPT-3.5-Turbo | 0-shot | 37.40 | 47.02 | 31.25 | 26.27 | 12.71 | 70.66 |
| | 1-shot | 40.76 | 53.57 | 12.50 | 27.54 | 26.23 | 86.10 |
| | 0-shot-CoT | 38.17 | 45.24 | 33.59 | 25.85 | 13.93 | 72.20 |
| | 1-shot-CoT | 39.64 | 51.79 | 10.94 | 26.70 | 26.23 | 84.56 |
| | 1-shot+DK | 49.36 | 62.50 | 35.55 | 38.98 | 26.23 | 85.71 |
| GPT-4 | 0-shot | 42.39 | 66.67 | 20.70 | 45.76 | 2.46 | 82.63 |
| | 1-shot | 47.98 | 67.86 | 26.56 | 44.07 | 14.75 | 91.12 |
| | 0-shot-CoT | 43.34 | 67.86 | 23.44 | 46.19 | 1.64 | 83.78 |
| | 1-shot-CoT | 47.46 | 67.26 | 30.08 | 41.95 | 11.07 | 91.12 |
| | 1-shot+DK | 53.22 | 64.88 | 43.75 | 49.58 | 20.08 | 89.56 |
| GPT-4o | 0-shot | 44.23 | 62.50 | 19.53 | 25.00 | 31.56 | 86.49 |
| | 1-shot | 49.36 | **69.05** | 26.56 | 30.93 | 34.43 | 89.97 |
| | 0-shot-CoT | 44.71 | 63.10 | 20.70 | 24.58 | 32.38 | 86.49 |
| | 1-shot-CoT | 48.67 | 67.86 | 25.78 | 28.81 | 32.79 | **91.89** |
| | 1-shot+DK | **64.83** | 61.31 | 65.23 | **59.32** | **46.31** | 89.19 |
| *Fine-tuned LLMs* | | | | | | | |
| SFT LLaMA-2 7B | 0-shot | 66.72 | 69.05 | 35.94 | 83.48 | 54.51 | 91.89 |
| SFT LLaMA-3 8B | 0-shot | **77.13** | **79.76** | 65.23 | **88.56** | 55.33 | **97.30** |
| SFT LLaMA-3 8B Instruct | 0-shot | 75.92 | 69.64 | **68.75** | 85.17 | **57.38** | 96.14 |
| *Human experiments (On subset of mini-StatQA)* | | | | | | | |
| Human (Non-Stats) | Closed-book | 18.10 | 5.88 | 3.85 | 8.70 | 0.00 | 65.39 |
| | Open-book | 34.48 | **52.94** | 0.00 | 30.44 | 8.33 | 84.62 |
| Human (Stats) | Closed-book | 23.28 | 29.41 | 0.00 | 17.39 | 0.00 | 69.23 |
| | Open-book | **53.45** | 47.06 | **23.08** | **65.22** | **37.50** | **92.31** |

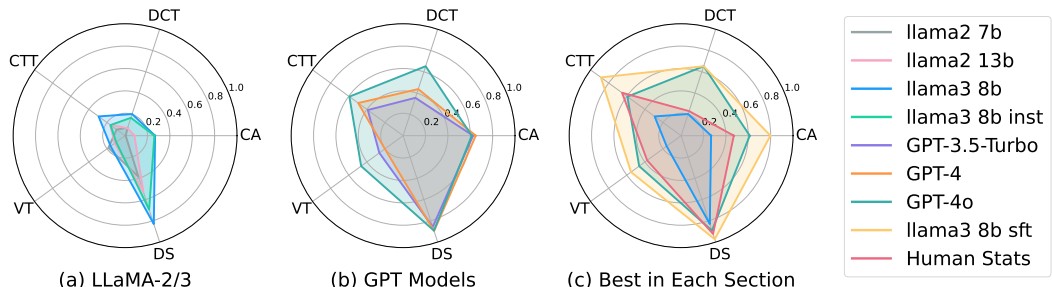

Figure 5: Best Results of Each Model in Five Sub-tasks (Section 2.1).

**Impact of Prompting Strategies.** We observe that compared to 0-shot, 1-shot learning increased the $Acc(\mathbb{C}, \mathbb{M})$ for the LLaMA-2/3 models by an average of 9.6%, and by 4.7% for GPT models. However, CoT leads to a minor performance decrease in most cases of LLaMA-2/3, whereas its impact on larger GPT models is negligible. More analysis for CoT is shown in Section E.2. The introduced domain knowledge prompting has proven effective, for six out of the seven non-fine-tuned LLMs obtained their best results in our experiments. Notably, it has a more pronounced effect on larger LLMs with stronger comprehension abilities, particularly GPT-4o.

> Finding 1. Few-shot learning and the inclusion of domain knowledge are helpful for LLMs in this task, whereas CoT is more likely to result in slight performance degradation in smaller models.

**Human Performance.** When completing the task independently without consulting any references, human participants demonstrate low accuracy. However, when provided with supplemental information of applicability, their performance improved significantly, particularly the participants in statistics, whose accuracy reached 53.4%, surpassing all non-fine-tuned LLMs with common prompting strategies. However, this is overshadowed by the fine-tuned models and the best performance of GPT-4o when domain knowledge is introduced in the prompt.

> Finding 2. LLMs with prompt-based approaches remain behind people in statistics. However, the gap can be filled even surpassed by fine-tuning or introducing domain knowledge to a strong LLM.

**Comparing LLMs and Human Performance.** LLMs and humans perform best on straightforward and commonly seen descriptive statistics tasks. In hypothesis testing tasks, humans and most LLMs exhibit relatively strong performances in correlation analysis and distribution compliance tests. In contrast, LLMs and humans encounter major challenges in contingency table tests and variance tests. For these tasks where performance is relatively weak, introducing domain knowledge to larger proprietary LLMs can yield significant improvements, especially GPT-4o, whereas improvements on smaller open-source LLMs are less conspicuous. This may be due to larger proprietary LLMs having inherently stronger comprehension abilities, enabling them to utilize domain knowledge more effectively to enhance performance, but open-source models are comparatively weaker in this regard.

> Finding 3. Humans and most LLMs are adept at descriptive statistics tasks but struggle with contingency tables and variance tests. Domain knowledge significantly boosts larger proprietary LLMs' performance, notably GPT-4o, but has limited impact on smaller open-source models.

## 3.3 Errors Analysis

**Errors Taxonomy.** We categorize errors into four distinct types and mixed errors, with examples in Section E.1. The error categories are (1) Invalid Answer: meaningless responses or do not conform to the required format; (2) Column Selection Error: irrelevant or incorrect column selection; (3) Statistical Task Confusion: confusion regarding the category of statistical tasks, leading to incorrect selection of methods; (4) Applicability Error: no confusion on task category but failing to discern the usage scenarios and prerequisites, resulting in the selection of inapplicable methods; (5) Mixed Errors: valid answer but contain multiple types of errors.

**Errors Analysis.** We examine and summarize the proportions and distributions of the error types across different LLMs and experimental setups, as shown in Figure 6. We observe that except smaller

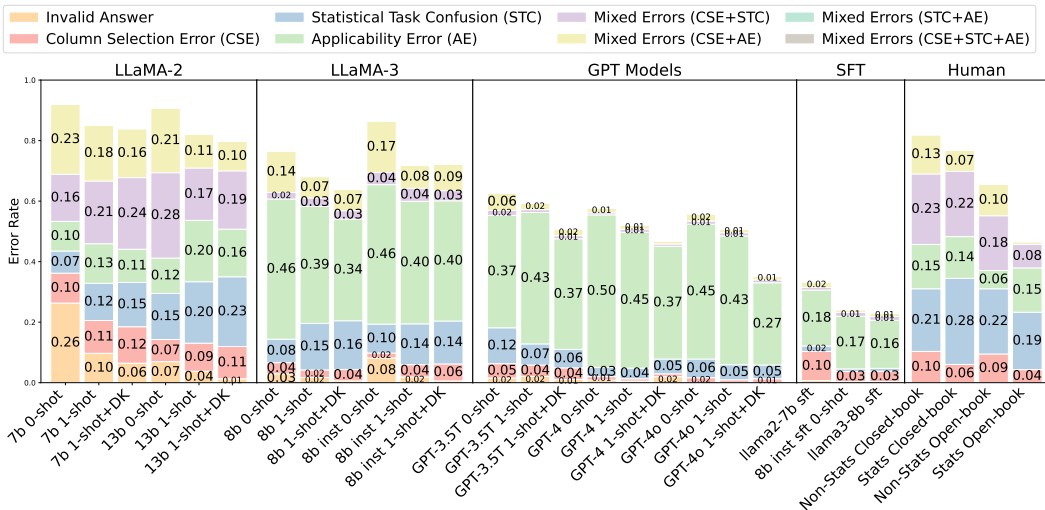

Figure 6: Distribution of Error Categories Across Experiments

models like LLaMA-2-7b fail to understand the task more frequently, resulting in invalid responses, others seldom respond with invalid answers. Additionally, column selection errors account for a non-negligible proportion of the LLaMA-2/3 but are not a barrier for the GPT models.

**Finding 4.** LLaMA-3 and GPT models demonstrate a competent understanding of tasks, and the latter can accurately select data columns, but LLaMA-2 models have difficulties in these aspects.

Except for LLaMA-2, the primary error type observed for LLMs on our benchmark is applicability errors, while the percentage of errors associated with statistical task confusion is quite low. Strongly performed GPT and fine-tuned models also have a considerable proportion of applicability errors, even if we provide domain knowledge of applicability. This can be considered an inherent limitation of LLMs, indicating a deficiency in LLMs' ability to understand and assess methodological applicability. However, the situation is completely different for humans. Although participants in statistics outperform all non-fine-tuned LLMs in open-book experiments, in all human experimental groups, applicability errors account for a smaller proportion, while statistical task confusion errors constitute the highest proportion.

**Finding 5.** LLMs are good at distinguishing different statistical tasks and then selecting associated methods, but they struggle to utilize domain knowledge to assess method applicability effectively. Conversely, humans excel at discerning method applicability but are prone to task confusion.

**Finding 6.** Humans and LLMs have distinct proficiencies and weaknesses in different aspects of selecting applicable statistical tasks, highlighting the potential for complementary collaboration.

## 4 Research Opportunity

**LLMs for Statistical Applications.** Current LLMs struggle with accurately assessing the applicability of statistical methods, even when provided with explicit domain knowledge. This indicates a profound need for developing models that can better understand and utilize detailed methodological prerequisites and application contexts. Future research should focus on integrating more sophisticated reasoning mechanisms into LLMs or leveraging a multi-agent framework to enhance their comprehension and application of statistical methods.

**Human-AI Collaboration in the Statistical Task.** LLMs and humans exhibit distinct aspects of superior capability in statistical tasks. Therefore, an in-depth study into harnessing their unique strengths for complementary collaboration to achieve optimal performance can be a valuable endeavor. This approach could leverage the computational efficiency and data handling capabilities of LLMs alongside the nuanced understanding and domain knowledge of human experts, leading to more robust and accurate statistical analysis.

**Expanding the Benchmark Dataset.** We establish `StatQA`, focusing on the evaluation of applicability rather than computational results, aiming to highlight the importance of discerning statistical method suitability. `StatQA` can be expanded to encompass a broader range of statistical tasks and methods, as discussed in Section G in the Appendix. Beyond the field of statistics, the evaluation of method applicability is also crucial in professional domains such as finance and operations research. Consequently, curating a more extensive benchmark represents a significant undertaking.

## 5 Related Work

**Large Language Models.** Proprietary LLMs like ChatGPT [27] and GPT-4 [28] exhibit impressive capabilities in text comprehension and processing, and recent-released GPT-4o [29] is further elevated performance and speed. Open-source models are also favored for their flexibility and suitability for customization, among the most representative examples are the LLaMA series models, such as LLaMA-2 [25] and the new LLaMA-3 [26]. Meanwhile, prompting strategies [32, 33] and fine-tuning methods like LoRA [31], enable LLMs to better adapt to specific tasks with less computational costs.

**Relevant Benchmarks.** Some benchmarks have been proposed to assess LLMs' problem-solving ability related to mathematical domains [34–38], but most are more focus on results reasoning and calculation instead of methods selection. Notable works like MATH [39] and GHOSTS [40] cover mathematical problems of varying difficulty, ranging from elementary to graduate levels; TheoremQA [41] and SciBench [42] involve multi-disciplinary problems; and recent studies like MathVista [14] and MathVerse [15] include visual tasks. DAEval [17] involves correlation analysis and distribution analysis as components of question concepts, but its coverage is relatively limited, and QRData [16] is for quantitative reasoning with data, but still more focus on calculation. Studies on datasets for more specialized statistical scenarios, particularly involving assessing the applicability of statistical methods, are minimal.

We also include a more detailed discussion about the related work in Section A in the Appendix.

## 6 Conclusion

In this paper, we propose `StatQA`, a benchmark designed for statistical analysis tasks. To evaluate the capabilities of LLMs on `StatQA`, we conduct systematic experiments with both open-source and closed-source LLMs to determine whether they can select proper statistical methods and relevant data columns by discerning prerequisites and assessing applicability, akin to competent statisticians. Furthermore, we conduct human experiments for a comparative study, discussing how humans and LLMs differ in their capabilities on `StatQA` and revealing their potential complementarity. Our findings suggest that while LLMs show promise, there is still significant room for improvement, especially in their ability to accurately assess the applicability of statistical methods. Future work could focus on enhancing the reasoning mechanisms of LLMs and exploring more effective human-AI collaboration strategies. We believe that `StatQA` fills a significant gap and provides a valuable resource for developing more advanced LLMs for statistical analysis tasks.

## 7 Acknowledgement

This paper is supported by NSF of China (62402409), Guangzhou Municipality Big Data Intelligence Key Lab (2023A03J0012), Guangdong Basic and Applied Basic Research Foundation (2023A1515110545), CCF-Huawei Populus Grove Fund (CCF-HuaweiDB202403), and the Red Bird Grant, Research Grant from The Hong Kong University of Science and Technology (Guangzhou).

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

# Appendices and Supplemental Materials

## A  Detailed Related Work

### A.1  Large Language Models (LLMs)

LLMs are increasingly employed in both academic and industrial sectors, with extensive applications across disciplines such as science, education, medicine, and finance [43–49]. Particularly since the advent of ChatGPT [27], LLMs have demonstrated impressive capabilities and versatility in data analysis, question answering, and reasoning [50–58]. The GPT-series model released by OpenAI is one of the most representative and widely used proprietary LLMs, including the newly released GPT-4o as well as ChatGPT and GPT-4, which have achieved state-of-the-art performance on numerous professional benchmarks without fine-tuning [27–29, 59]. Other renowned proprietary LLMs include Claude [60], Qwen [61], and Gemini [62].

Open-source LLMs gain significant attention and favor by virtue of their openness, flexibility, and suitability for task-specific fine-tuning. Notably, Meta's LLaMA series, including LLaMA-2 [25] and the new LLaMA-3 [26] are representative examples. Additionally, techniques such as in-context learning [30, 32] and fine-tuning approaches like LoRA [31], facilitate the enhancement of LLMs' performance in specific tasks with limited computational costs.

### A.2  Relevant Benchmarks

Some benchmarks have been established to assess LLMs' problem-solving and reasoning abilities within scientific domains in recent years. For multi-disciplinary scientific question-answering datasets, TheoremQA [41] is introduced to evaluate LLMs' capacity to apply theorems in solving complex scientific problems, covering 350 theorems across mathematics, physics, EE&CS, and finance. SciBench [42] focuses on vision context-involved scientific problem-solving in collegiate-level mathematics, physics, and chemistry.

There are notable works that have concentrated on mathematical reasoning [36], addressing various types of problems and levels of difficulty, ranging from grade school to postgraduate level. GSM8K [63] is a comprehensive dataset of grade-level math word problems; however, with the rapid development of LLMs, GSM8K is no longer sufficiently challenging for proprietary LLMs now. MATH [39] serves as a benchmark comprised of multiple-choice problems covering a range of difficulty levels from elementary school to college. In contrast, Mishra et al. introduce LILA [37], a sophisticated benchmark encompassing diverse question formats and higher difficulty levels, including problems in calculus and algebra. Following the attention garnered by ChatGPT's remarkable performance, the more advanced GHOSTS benchmark [40] has been proposed to evaluate the capabilities of ChatGPT and GPT-4 on graduate-level mathematics problems. Recent studies like MathVista [14] and MathVerse [15] systematically evaluate LLMs' performance in visual math problem-solving.

Regarding benchmarks involved in statistical tasks, DAEval [17] involves correlation analysis and distribution analysis as components of question concepts, and QRData [16] is designed to assess the quantitative reasoning capabilities of LLMs using real-world data, including statistical and causal

reasoning. Nevertheless, the emphasis of these benchmarks is on the accuracy of computational outcomes, with relatively limited attention to the breadth of statistical tasks covered. Research on datasets tailored to more specialized statistical scenarios, particularly those evaluating the applicability of statistical methods, remains sparse.

# B   More Details of StatQA

## B.1   Access to StatQA

`StatQA` contains 11,623 examples and mini-`StatQA` contains 1,163 examples across five typical categories of statistical tasks. To facilitate further research, we make our code and data available at `https://statqa.github.io/`, under GPL-3 license.

## B.2   Statistical Tasks and Associated Methods

The five typical categories of statistical tasks and their associated methods covered in this paper are listed in Table 3.

Table 3: Table of Covered Statistical Task Categories and Methods in `StatQA`.

| Category | Description | Methods |
|---|---|---|
| Correlation Analysis | Assess the strength and direction of the correlation. | Pearson Correlation Coefficient
Spearman Correlation Coefficient
Kendall Correlation Coefficient
Partial Correlation Coefficient |
| Contingency Table Test | Evaluate the independence between categorical variables. | Chi-square Independence Test
Mantel-Haenszel Test
Fisher Exact Test |
| Distribution Compliance Test | How well a data set conforms to a specified distribution. | Anderson–Darling Test
Lilliefors Test
Shapiro-Wilk Test of Normality
Kolmogorov-Smirnov Test for Normality
Kolmogorov-Smirnov Test for Uniform distribution
Kolmogorov-Smirnov Test for Gamma distribution
Kolmogorov-Smirnov Test for Exponential distribution
Kolmogorov-Smirnov Test (Distributions Comparison) |
| Variance Test | Compare the variability of groups to assess statistical differences | F-Test for Variance
Mood Variance Test
Levene Test
Bartlett Test |
| Descriptive Statistics | Summarize and describe data features. | Mean, Median, Mode, Range, Quartile,
Standard Deviation, Skewness, Kurtosis |

## B.3   Examples of Question Templates

As mentioned in Section 2.3, we employ question templates $T$ to synthesize statistical questions in preliminary dataset. These templates $T$ include placeholders where relevant data columns to be involved, and Table 4 presents some examples of question templates across various categories.

## B.4   More Examples in StatQA

Figure 7 presents more examples in `StatQA` across five task categories.

## B.5   Domains of Source Tables

As mentioned, we have assembled 78 tables to serve as the foundation for the construction of `StatQA`, encompassing six domains in real-world scenarios: education, medicine, science, engineering, economy, and life. In the process of dataset partitioning (**Step 4** in Section 2.3), we ensure comprehensive representation across all domains when dividing the training and test sets derived from these source tables. The distribution of source tables utilized for testing (to establish `StatQA` for the evaluation) and for training (to develop $\mathbb{D}_{\text{train}}$ for fine-tuning) is detailed in Table 5.

Table 4: Examples of statistical question templates used in `StatQA` construction.

| Category | Question Template Examples |
|---|---|
| Correlation Analysis | How strong is the correlation between {Column 1} and {Column 2}? 
 Is there a linear relationship between {Column 1} and {Column 2}? 
 When {Control Column} is fixed, how do {Column 1} and {Column 2} correlate? |
| Contingency Table Test | Is {Column 1} independent of {Column 2}? 
 How are the categories of {Column 1} reflected in the frequency of {Column 2}? 
 Is there a significant difference in the impact of {Column 1} on {Column 2} across different levels of {Strata Column}? |
| Distribution Compliance Test | Does {Column} follow a normal distribution? 
 Is the {Distribution} distribution a good fit for {Column} data? 
 Do {Column 1} and {Column 2} follow a similar distribution curve? |
| Variance Test | Can we assume there is no major difference in variances for {Column 1} and {Column 2}? 
 Is the degree of spread in {Column 1} similar to that in {Column 2}? 
 Are the variances in {Column 1} and {Column 2} significantly different? |
| Descriptive Statistics | What is the average of the {Column} values? 
 What's the most common value in {Column}? 
 How wide is the range of values in {Column}? 
 Can you calculate the quartiles for {Column}? 
 What is the measure of standard variability for {Column}? 
 How does the asymmetry of {Column}? 
 How peaked is the distribution of {Column}? |

Figure 7: More Examples in `StatQA`.

# C More Details of Experimental Setups

## C.1 Hyperparameter and Hardware Platfrom

In our evaluations, we set the hyperparameter temperature=0 for all LLMs and top_p=1.0 for all LLaMA-2/3 models to reduce randomness. We use the vLLM framework [64] to conduct evaluations for LLaMA-2/3 models on 8 NVIDIA 4090 and call OpenAI API to obtain answers generated by GPT models. Differently, the GPT-3.5-Turbo's temperature is set to 0.7 in statistical question refinement, for more diverse and flexible expressions.

Table 5: Domains Distribution of Source Tables

| Tables' Domain | `StatQA` (for test) | $\mathbb{D}_{train}$ (for training) |
|---|---|---|
| Education | 9 | 5 |
| Medicine | 4 | 7 |
| Science | 4 | 3 |
| Engineering | 3 | 4 |
| Economy | 10 | 13 |
| Life | 7 | 9 |

In fine-tuning, we utilize the LLaMA-Factory framework [65] to fine-tune LLaMA-2-7b-chat-hf, Meta-Llama-3-8B, and Meta-Llama-3-8B-Instruct on an NVIDIA A800 (80G), for a total of 3 epochs on the training set, set the batch size to 16, and use the Adam optimizer. The learning rate linearly increases from 0 to 5e-5 in the first 200 steps of fine-tuning, then gradually decreases to 0 at the end using the cosine annealing strategy.

## C.2 More Details of Human Experiment Setups

We develop a testing platform for human experiments to facilitate the operations of participants. The GUI of the testing platform is shown in Figure 8. The examples for human experiments are substantially evenly divided into task blocks based on task category and difficulty, for closed-book test and open-book test respectively. Each participant needs to complete two blocks, using different answering modes (closed-book first and then open-book) for each block. Consistent with LLMs' experiments, participants must complete the task block for the closed-book test independently without consulting any external references, and domain knowledge in Table 9 will also be provided to participants in open-book human experiments. Participants are compensated about $8 per hour and about $32 in total for each participant.

To ensure all participants clearly understand our tasks as well as the requirements of answering modes in experiments and are familiar with operations for the testing platform, we perform all human experiments as following procedures: (1) Introduction to the task and requirements by the organizers; (2) Participants watch the tutorial video of the testing platform; (3) Distribute accounts and passwords to participants, who will then log into the platform for corresponding experiments.

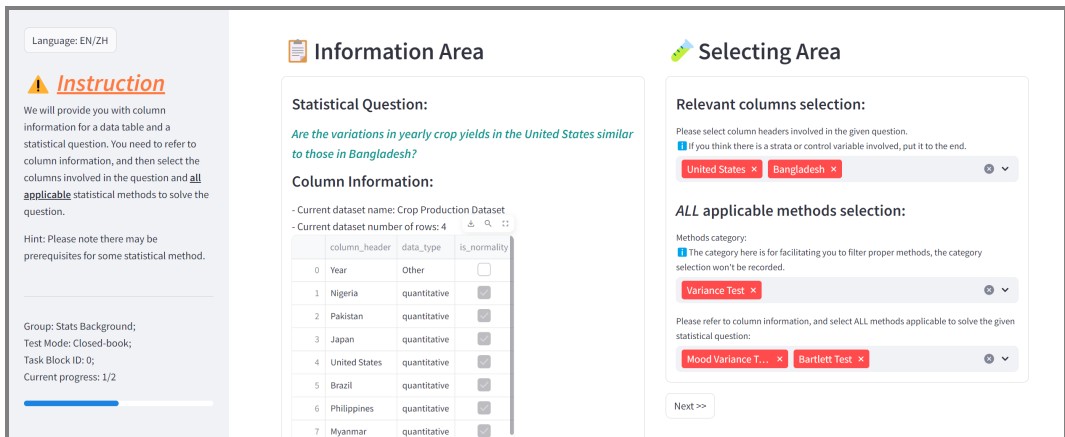

Figure 8: GUI of the Testing Platform for Human Experiments.

## C.3 Prompts

The prompt used for refining the phrasing and expression of statistical questions in the `StatQA` is shown in Table 6. The prompts used to generate responses from LLMs is shown in Table 7, including the following components: task description, instruction, classification list, demonstration example (optional), column information, and statistical question. Table 8 illustrates examples used in one-shot learning and one-shot-CoT respectively. Table 9 demonstrates the introduction of domain knowledge (DK) in prompt structure.

Table 6: Prompt for Statistical Question Refinement. Input origin question sentences to be refined, and descriptive information (if any), instruct GPT-3.5-Turbo to refine and paraphrase for more coherent and diverse expressions without changing the original meaning.

| Prompt for LLM Statistical Question Refinement |
| --- |
| SYSTEM:
I'm a native English-speaking statistician. I will help you refine and improve expressions of statistical sentences without changing the original meaning. Please tell me the sentence you want to refine. |
| USER:
Suppose you're a native English-speaking statistician, and I will give you a sentence about a statistical problem. You need to improve the English expression of the given sentence to make it grammatically and semantically correct, statistically rigorous and more coherent in expression. The given sentence will contain the names of the variables to be analyzed, and you are encouraged to based on the description, change them to more natural expressions without affecting the meaning. You can be flexible in how you improve the expression, but you must not change the original meaning of the sentence.
Variable description (Optional):
...... Description for involved variables (columns)
Sentence:
...... Preliminary statistical question to be refined |

# D   More Experimental Results

To accommodate users only accessible to limited computational resources, or those with sufficient resources but are concerned about data privacy, a more extensive evaluation of open-source LLMs is essential. Accordingly, to provide references for these two groups of users, we conduct additional experiments on both smaller and larger representative open-source models, including Qwen2-0.5B-Instruct, Qwen2-1.5B-Instruct, Qwen2-72B-Instruct, Meta-Llama-3-8B, and Yi-34B-Chat. Table 10 illustrates experimental results for additional evaluation on open-source LLMs.

Qwen2-0.5B-Instruct is entirely inadequate for our task, presenting only 2.41% 1-shot overall accuracy and extremely low capabilities in all tasks. Qwen2-1.5B-Instruct demonstrates performance levels comparable to certain LLaMA models, specifically falling between LLaMA-2-13b and LLaMA-3-8b. Given that Qwen2-1.5B-Instruct is significantly smaller in parameters, although its performance is still not robust, it represents noteworthy and encouraging results for resource-constrained users.

In contrast, larger open-source LLMs exhibit significant enhancements, underscoring the importance of model scale. Remarkably, Qwen2-72B-Instruct demonstrates impressive performance with a 47.12% one-shot overall accuracy, surpassing all other open-source LLMs and being comparable to GPT models. Its strong capabilities are particularly promising for users with ample resources who prioritize data privacy and seek to minimize the reliance on proprietary solutions.

Table 10: More Open-source LLMs Experimental Results of $Acc(\mathbb{C}, \mathbb{M})$ (%) on mini-StatQA. The 1st , 2nd , 3rd place results in all additional experiments here are highlighted in red, blue, and green respectively. The **bold** results are the best in each section.

| Model | Strategy | Overall | CA | CTT | DCT | VT | DS |
| --- | --- | --- | --- | --- | --- | --- | --- |
| Smaller Open-source LLMs | | | | | | | |
| Qwen2 0.5B Instruct | 0-shot | 1.38 | 0.00 | 0.00 | 1.27 | 0.00 | 5.02 |
| | 1-shot | 2.41 | 0.00 | 0.00 | 3.81 | 1.23 | 6.18 |
| Qwen2 1.5B Instruct | 0-shot | 15.99 | 0.00 | 1.56 | 1.27 | 18.03 | 52.12 |
| | 1-shot | **20.38** | **0.60** | **7.42** | **4.24** | **22.54** | **58.69** |
| Larger Open-source LLMs | | | | | | | |
| Qwen2 72B Instruct | 0-shot | 44.71 | 57.74 | **30.47** | **36.02** | 10.66 | 90.35 |
| | 1-shot | **47.12** | **66.07** | 30.47 | 28.81 | 21.72 | **91.89** |
| LLaMA-3 70B | 0-shot | 27.09 | 35.12 | 2.34 | 11.02 | 19.67 | 67.95 |
| | 1-shot | 34.14 | 50.60 | 9.38 | 14.83 | 24.59 | 74.52 |
| Yi 34B Chat | 0-shot | 28.03 | 41.07 | 13.67 | 14.41 | 0.00 | 72.59 |
| | 1-shot | 33.96 | 45.24 | 19.14 | 24.15 | 10.25 | 72.59 |

Table 7: Prompt Structure for Answer Generation in LLMs' Experiments. For zero-shot learning, the part of the demonstration example will not be included.

| **General Prompt Structure for LLM Answer Generation** |
| --- |
| ### Task Description: 
 You need to select relevant columns and all applicable methods from provided list for the given statistical question. |
| ### Instruction: 
 You should only reply with one answer in JSON format containing two keys: "columns" and "methods". The value of "columns" is a list of columns' headers relevant to the given statistical question, and the value of "methods" is a list containing all methods you think applicable. For example: "columns": ["c1", "c2", "..."], "methods": ["m1", "m2", "..."]. If you think there is a strata or control variable involved, put its column header in the last item of the columns list. Ensure your methods selection is only limited to the classification list provided. |
| ### Classification List: 
 Correlation Analysis: Pearson Correlation Coefficient, Spearman Correlation Coefficient, Kendall Correlation Coefficient, Partial Correlation Coefficient; 
 Distribution Compliance Test: Anderson-Darling Test, Shapiro-Wilk Test of Normality, Kolmogorov-Smirnov Test for Normality, Lilliefors Test, Kolmogorov-Smirnov Test, Kolmogorov-Smirnov Test for Uniform distribution, Kolmogorov-Smirnov Test for Gamma distribution, Kolmogorov-Smirnov Test for Exponential distribution; 
 Contingency Table Test: Chi-square Independence Test, Fisher Exact Test, Mantel-Haenszel Test; 
 Descriptive Statistics: Mean, Median, Mode, Range, Quartile, Standard Deviation, Skewness, Kurtosis; 
 Variance Test: Mood Variance Test, Levene Test, Bartlett Test, F-Test for Variance. |
| ### Demonstration Example: (Optional) 
 ...... Example for few-shot learning, as shown in Table 8 
 <example start> 
 # Column Information: 
 ...... 
 # Statistical Question: 
 ...... 
 # Correct Answer: 
 ...... 
 </example end> |
| ### Column Information: 
 ...... Column information of the current source tabular data |
| ### Statistical Question: 
 ...... Current statistical question |
| ### Response: 
 For zero-shot-CoT experiments: 
 Let's work this out in a step-by-step way to be sure we have the right answer. 
 General: 
 The answer of relevant columns and applicable methods in JSON format is: |

Table 8: Examples Used in Few-shot Learning and CoT Respectively. In accordance with Section 3.1.1, we have design five examples corresponding to five task categories, for random selection.

| **An Example Used in One-shot Learning** |
| --- |
| # Column Information:
column_header: TV Ad Budget ($); data_type: quantitative; num_of_rows: 200; is_normality: False.
column_header: Radio Ad Budget ($); data_type: quantitative; num_of_rows: 200; is_normality: False.
column_header: Newspaper Ad Budget ($); data_type: quantitative; num_of_rows: 200; is_normality: False.
column_header: Sales ($); data_type: quantitative; num_of_rows: 200; is_normality: False.
# Statistical Question: Is there a linear correlation between the TV advertising budget ($) and sales revenue ($) in this study?
# Correct Answer: {"columns": ["TV Ad Budget ($)", "Sales ($)"], "methods": ["Pearson Correlation Coefficient", "Spearman Correlation Coefficient", "Kendall Correlation Coefficient"]} |

| **An Example Used in One-shot-CoT** |
| --- |
| # Column Information:
column_header: TV Ad Budget ($); data_type: quantitative; num_of_rows: 200; is_normality: False.
column_header: Radio Ad Budget ($); data_type: quantitative; num_of_rows: 200; is_normality: False.
column_header: Newspaper Ad Budget ($); data_type: quantitative; num_of_rows: 200; is_normality: False.
column_header: Sales ($); data_type: quantitative; num_of_rows: 200; is_normality: False.
# Statistical Question: Is there a linear correlation between the TV advertising budget ($) and sales revenue ($) in this study?
# Correct Answer: Firstly, the question asks about TV advertising budget and sales revenue, so the relevant columns are TV Ad Budget ($) and Sales ($). Secondly, given that it asks about correlation, and the variables involved are all quantitative, some methods from Correlation Analysis can be applicable. Thirdly, there are enough samples of 200, with only two variables involved, and no control variables, so the applicable methods are Pearson Correlation Coefficient, Spearman Correlation Coefficient, Kendall Correlation Coefficient. Hence, the answer is: {"columns": ["TV Ad Budget ($)", "Sales ($)"], "methods": ["Pearson Correlation Coefficient", "Spearman Correlation Coefficient", "Kendall Correlation Coefficient"]} |

Table 9: Introduce Domain Knowledge (DK) in the Prompt. Based on one-shot learning prompt structure and replace the "Classification List" part with the contents in this table. Domain knowledge shown in this table is also provided to participants in open-book human experiments for consistency.

| **Introduce Domain Knowledge (DK) in Prompt** |
| --- |
| ### Methods and applicable usage scenarios:
# Correlation Analysis
Pearson Correlation Coefficient, Spearman Correlation Coefficient: Correlation analysis for two quantitative variables;
Kendall Correlation Coefficient: Correlation analysis for two quantitative variables, suitable for small samples;
Partial Correlation Coefficient: Correlation analysis when involving controlling variable;
# Distribution Compliance Test
Anderson-Darling Test, Kolmogorov-Smirnov Test for Normality: Test for normality;
Shapiro-Wilk Test of Normality: Test for normality, suitable for small samples;
Lilliefors Test: Test for normality, suitable for large samples;
Kolmogorov-Smirnov Test: Comparison of distribution between two independent samples;
Kolmogorov-Smirnov Test for Uniform distribution, Kolmogorov-Smirnov Test for Gamma distribution, Kolmogorov-Smirnov Test for Exponential distribution: Test for corresponding distributions;
# Contingency Table Test
Chi-square Independence Test: Contingency table test of large sample categorical variables;
Fisher Exact Test: Contingency table test of small sample categorical variables;
Mantel-Haenszel Test: Contingency table test when strata data to be controlled;
# Variance Test
Mood Variance Test, Levene Test: Whether there is a significant difference;
Bartlett Test, F-Test for Variance: Whether there is a significant difference in variance between normally distributed variables;
# Descriptive Statistics
Mean, Median, Mode, Range, Quartile, Standard Deviation, Skewness, Kurtosis. |

# E More Error Analysis

## E.1 Examples of Error Taxonomy

In Section 3.3, we conduct the error analysis, identifying the following types of errors: invalid answers, column selection errors, statistical task confusion, applicability errors, and mixed errors. Table 11 provides examples for each category within this error taxonomy.

Table 11: Examples for Different Error Types. The red texts indicate the incorrect parts.

| Examples for different error types |
|---|
| *Invalid Answer* |
| **Statistical Question:** Is the Exponential distribution a suitable model for representing the distribution of Grade Point Average (GPA) of the applicants during their undergraduate studies? |
| **Ground Truth:** {"columns": ["CGPA"], "methods": ["Kolmogorov-Smirnov Test for Exponential distribution"]} |
| **Model Answer:** Please select the relevant columns and applicable methods for the given statistical question. Please select the relevant columns and applicable methods for the given statistical question... |
| *Column Selection Error* |
| **Statistical Question:** Is the variability in GRE scores not significantly different from that in Letter of Recommendation? |
| **Ground Truth:** {"columns": ["GRE Score", "LOR"], "methods": ["Mood Variance Test", "Levene Test"]} |
| **Model Answer:** {"columns": ["CGPA", "LOR"], "methods": ["Mood Variance Test", "Levene Test"]} |
| *Statistical Task Confusion* |
| **Statistical Question:** Is there a relationship between the sex of the patient and the type of chest pain experienced? |
| **Ground Truth:** {"columns": ["Sex", "ChestPainType"], "methods": ["Chi-square Independence Test"]} |
| **Model Answer:** {"columns": ["Sex", "ChestPainType"], "methods": ["Pearson Correlation Coefficient", "Spearman Correlation Coefficient", "Kendall Correlation Coefficient"]} |
| *Applicability Error* |
| **Statistical Question:** Is there a statistically significant difference between the variances in height and weight of the Pokemon? |
| **Ground Truth:** {"columns": ["Height_m", "Weight_kg"], "methods": ["Mood Variance Test", "Levene Test"]} |
| **Model Answer:** {"columns": ["Height_m", "Weight_kg"], "methods": ["Mood Variance Test", "Levene Test", "Bartlett Test", "F-Test for Variance"]} |
| *Mixed Error* |
| **Statistical Question:** Does the consumption of food between meals have any connection to the presence of family history with overweight? |
| **Ground Truth:** {"columns": ["family_history_with_overweight", "CAEC"], "methods": ["Chi-square Independence Test"]} |
| **Model Answer:** {"columns": ["FAVC", "family_history_with_overweight"], "methods": ["Chi-square Independence Test", "Fisher Exact Test", "Mantel-Haenszel Test"]} |

Explanations and remarks for examples in Table 11:

- **Invalid Answer:** fail to understand tasks, smaller models with weaker text comprehension capability such as LLaMA-2-7b are more likely to suffer from this problem;
- **Column Selection Error:** misselection of relevant columns due to failure to correctly understand the meaning: in this case of asking about the student's GRE scores and letters of recommendation, the model incorrectly chooses the irrelevant student's CGPA;
- **Statistical Task Confusion:** the sex and type of chest pain in the dataset are categorical, and the relationship between them should be calculated using methods in the contingency table test rather than correlation analysis;
- **Applicability Error:** correctly determine the task type as a variance test. However, the provided column information clearly shows that "Height_m" and "Weight_kg" do not follow a normal distribution, so the Bartlett Test and F-Test for Variance methods are not applicable. In this case, the model failed to discern the prerequisites leading to misselection of inapplicable methods.

Table 12: $\Delta Acc(\mathbb{C}, \mathbb{M})$ (%) After Introducing CoT Strategy in Different Experiments. Positive number indicates an `increase` in $Acc(\mathbb{C}, \mathbb{M})$ compare with same prompting strategy without CoT, and negative number indicates a performance `decrease` after introducing CoT.

| Model | Strategy | Overall | CA | CTT | DCT | VT | DS |
|---|---|---|---|---|---|---|---|
| Open-source LLMs: LLaMA-2/3 | | | | | | | |
| LLaMA-2 7B | 0-shot-CoT | -1.72 | -0.60 | -0.39 | 0.00 | -1.23 | -5.79 |
| | 1-shot-CoT | -0.52 | 1.19 | -1.95 | 2.54 | 0.41 | -3.86 |
| LLaMA-2 13B | 0-shot-CoT | -0.26 | 0.60 | -0.39 | 0.85 | -0.41 | -1.54 |
| | 1-shot-CoT | -0.34 | -2.98 | 3.91 | 0.42 | -1.64 | -2.32 |
| LLaMA-3 8B | 0-shot-CoT | -1.55 | 0.00 | 0.39 | -0.85 | -2.46 | -4.25 |
| | 1-shot-CoT | 0.34 | -3.57 | -2.34 | 1.27 | 3.69 | 1.55 |
| LLaMA-3 8B Instruct | 0-shot-CoT | -2.06 | 0.60 | 1.56 | 1.70 | -1.23 | -11.58 |
| | 1-shot-CoT | 0.09 | -0.60 | 4.69 | 2.54 | 0.00 | -6.18 |
| Proprietary LLMs: ChatGPT, GPT-4 and GPT-4o | | | | | | | |
| ChatGPT | 0-shot-CoT | 0.77 | -1.79 | 2.34 | -0.42 | 1.23 | 1.55 |
| | 1-shot-CoT | -1.12 | -1.79 | -1.56 | -0.85 | 0.00 | -1.54 |
| GPT-4 | 0-shot-CoT | 0.95 | 1.19 | 2.74 | 0.42 | -0.82 | 1.16 |
| | 1-shot-CoT | -0.52 | -0.59 | 3.52 | -2.12 | -3.69 | 0.00 |
| GPT-4o | 0-shot-CoT | 0.43 | 0.59 | 1.17 | -0.42 | 0.82 | 0.00 |
| | 1-shot-CoT | -0.69 | -1.19 | -0.78 | -2.12 | -1.64 | 1.93 |

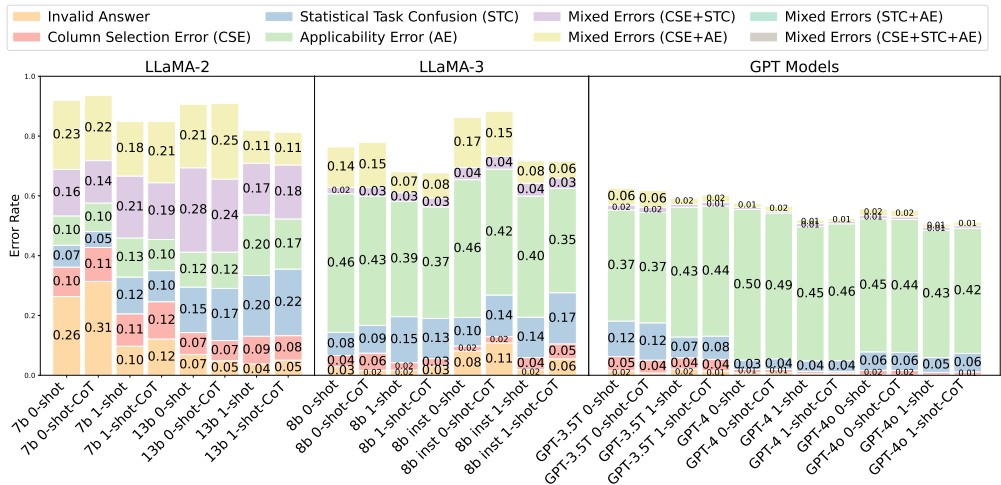

Figure 9: Distribution of Error Categories Across with/without CoT Experiments.

## E.2 Error Analysis for Introducing CoT

In Section 3.3, we mention that LLMs' overall performance can slightly decrease if CoT is introduced, especially in smaller LLMs. Table 12 illustrates the accuracy changes after introducing the CoT strategy in different experiments, noted as $\Delta Acc(\mathbb{C}, \mathbb{M})$, and Figure 9 shows the comparison of the error distributions on experiments with or without CoT.

As presented in Table 12, the 0-shot overall performance of smaller LLMs, namely LLaMA-2-7b, LLaMA-3-8b, and LLaMA-3-8b-Instruct, experiences a slight decline following the introduction of CoT. This performance degradation is most notable in descriptive statistics tasks. However, the negative impact of CoT on 1-shot learning experiments is mitigated, resulting in nearly identical overall performance between one-shot and one-shot-CoT experiments. In contrast, for larger models such as LLaMA-2-13b and GPT models, the influence of CoT is negligible. As illustrated in Figure 9, the introduction of CoT in smaller models including LLaMA-2-7b, LLaMA-3-8b, and LLaMA-3-8b-Instruct, tends to slightly increase the incidence of invalid answers and column selection errors, leading to a marginal decline in overall performance. However, the impact of CoT on error distribution in 1-shot experiments or on LLaMA-2-13b and GPT models is not prominent.

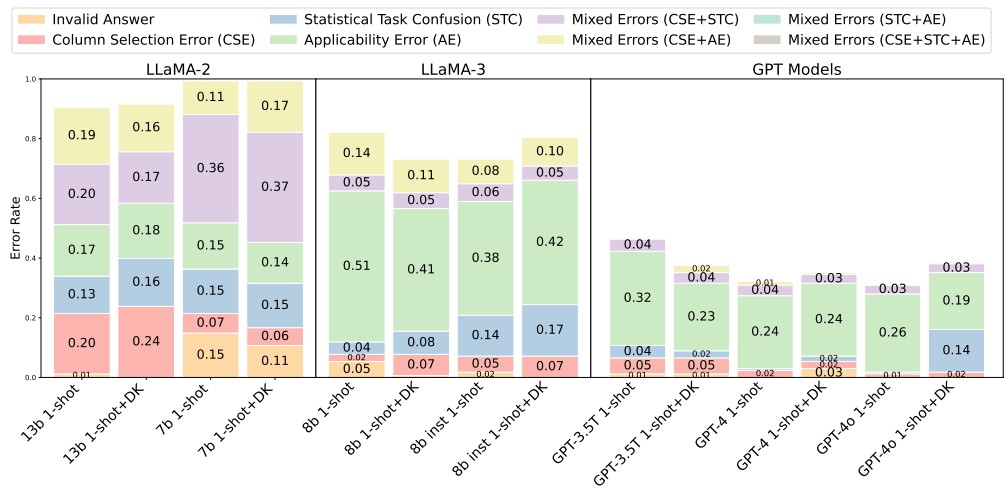

Figure 10: Distribution of Error Categories Across with/without DK on Correlation Analysis Task.

### E.3 Error Analysis for Introducing Domain Knowledge

As reported and analyzed in Section 3.2, the introduction of domain knowledge (DK) significantly boosts the overall performance of LLMs, particularly with larger proprietary LLMs like GPT models. However, as also shown in Table 2, it adversely affects some models' accuracy in correlation analysis (CA) and descriptive statistics (DS) tasks.

To further investigate, we also conduct an error analysis comparing 1-shot and 1-shot+DK approaches in the CA task, as depicted in Figure 10. It reveals a reduction in applicability errors across most LLMs, yet an increase in column selection errors and statistical task confusion among certain LLMs (*i.e.,* LLaMA-2-13b, LLaMA-3-8b-Instruct, GPT-4, GPT-4o) following the introduction of DK. Conversely, DK tends to positively influence the performance of more complex and less common tasks like DCT, CTT, and VT, thus contributing positively to overall performance.

We propose that the observed phenomenon may be attributed to the following potential factors: (1) Since DCT, CTT, and VT tasks are less commonly seen compared to CA and DS, LLMs are less involved in their pre-training data, making them relatively unfamiliar with these tasks, so the introduction of DK obtains substantial improvement; (2) For relatively simple and straightforward tasks (CA and DS), where LLMs possess some familiarity, the introduction of DK possibly leads to information obfuscation, resulting in varying levels of performance degradation on simpler subtasks. Deeper reasons may stem from various processes of LLMs' pre-training and the underlying mechanics of inference, which represents a valuable avenue for future research.

### E.4 Details of Statistical Task Confusion Error

We select two analogous experiments with domain knowledge provided to open-source and proprietary LLMs to compare the confusion matrices with open-book human experiments, shown in Figure 11. We observe that human participants most frequently confuse contingency table test tasks with correlation analysis tasks. This confusion is the primary factor contributing to their poor performance in contingency table test tasks. This issue arises from a lack of familiarity with the usage scenarios of contingency table tests resulting in difficulty in correctly differentiating task categories. Furthermore, both LLMs and humans occasionally confuse variance tests with correlation analysis tasks.

## F Qualitative Analysis

In this section, we revise the prompt, instructing LLMs to provide more detailed reasoning for their selections. We present typical examples aimed at qualitative analysis. Examples demonstrated include both success and failure, as well as other notable cases, along with an analysis of their answers and reasoning. Each example consists of the statistical question, column information, model answer, and ground truth. For brevity, we may show only partial column information in some instances.

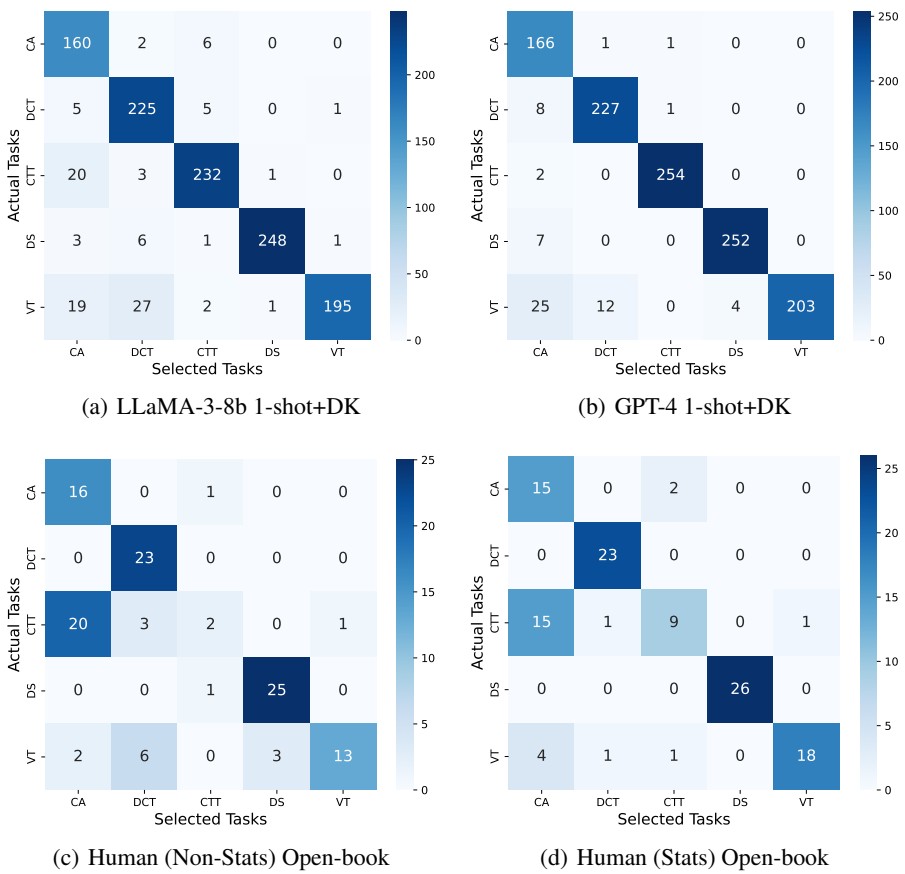

(a) LLaMA-3-8b 1-shot+DK

(b) GPT-4 1-shot+DK

(c) Human (Non-Stats) Open-book

(d) Human (Stats) Open-book

Figure 11: Comparison of Confusion Matrices on Statistical Tasks. (a) and (b): Confusion matrices of LLaMA-3-8b and GPT-4 with 1-shot+DK; (c) and (d): Confusion matrices of open-book experiments for non-statistical and statistical background participants respectively. The overall $Acc(\mathbb{C}, \mathbb{M})$ of these four experiments: (a) 36.11%, (b) 53.22%, (c) 34.48%, (d) 53.45%.

## F.1 Success Examples

Table 13: A Success Example. Correctly select the relevant columns, identify the application scenario of independence testing, check the prerequisites of categorical data type, and correctly select all applicable methods: Chi-square Independence Test and Fisher Exact Test, which belongs to Contingency Table Test.

| |
| --- |
| **Statistical Question: Do the observed frequencies of anxiety symptoms and alcohol consumption status conform to the independence assumption?** |
| **Column Information:** |
| column_header: GENDER; data_type: categorical; num_of_rows: 309; is_normality: False. |
| column_header: AGE; data_type: quantitative; num_of_rows: 309; is_normality: False. |
| column_header: SMOKING; data_type: categorical; num_of_rows: 309; is_normality: False. |
| column_header: ANXIETY; data_type: categorical; num_of_rows: 309; is_normality: False. |
| column_header: ALCOHOL CONSUMING; data_type: categorical; num_of_rows: 309; is_normality: False. |
| column_header: CHEST PAIN; data_type: categorical; num_of_rows: 309; is_normality: False. |
| column_header: LUNG_CANCER; data_type: categorical; num_of_rows: 309; is_normality: False. |
| ...... |
| **Model Answer:** |
| {"columns": ["ANXIETY", "ALCOHOL CONSUMING"],"methods": ["Chi-square Independence Test", "Fisher Exact Test"]} |
| Reason: The relevant columns for this statistical question are "ANXIETY" and "ALCOHOL CONSUMING" as they represent the variables of interest. The applicable methods for testing the independence assumption between these categorical variables are the Chi-square Independence Test and the Fisher Exact Test. |
| **Ground Truth:** |
| {"columns": ["ANXIETY", "ALCOHOL CONSUMING"], "methods": ["Chi-square Independence Test", "Fisher Exact Test"]} |

Table 14: A Success Example. Correctly select the relevant columns, identify the application scenario of variance test, check the prerequisites of normality, and correctly select all applicable methods.

| |
| --- |
| **Statistical Question: Are the levels of dispersion in crop production similar between Nigeria and Bangladesh?** |
| **Column Information:** |
| column_header: Year; data_type: Other; num_of_rows: 4; is_normality: False. |
| column_header: Nigeria; data_type: quantitative; num_of_rows: 4; is_normality: True. |
| column_header: Japan; data_type: quantitative; num_of_rows: 4; is_normality: True. |
| column_header: United States; data_type: quantitative; num_of_rows: 4; is_normality: True. |
| column_header: Indonesia; data_type: quantitative; num_of_rows: 4; is_normality: True. |
| column_header: Bangladesh; data_type: quantitative; num_of_rows: 4; is_normality: True. |
| column_header: China; data_type: quantitative; num_of_rows: 4; is_normality: True. |
| ...... |
| **Model Answer:** |
| {"columns": ["Nigeria", "Bangladesh"],"methods": ["Mood Variance Test", "Levene Test", "Bartlett Test", "F-Test for Variance"]} |
| Reason: The relevant columns for comparing the levels of dispersion in crop production between Nigeria and Bangladesh are "Nigeria" and "Bangladesh" as they contain quantitative data related to crop production. The applicable methods for this comparison include Mood Variance Test, Levene Test, Bartlett Test, and F-Test for Variance, which are used to determine if there is a significant difference in variance between the two normally distributed variables. |
| **Ground Truth:** |
| {"columns": ["Nigeria", "Bangladesh"],"methods": ["Mood Variance Test", "Levene Test", "Bartlett Test", "F-Test for Variance"]} |

## F.2 Failure Examples

Table 15: A Failure Example. The "Height_m", "Weight_kg" are not normally distributed according to provided column information, but the model hallucinates the normality leading to failure of discern applicable methods, misselecting "Bartlett Test" and "F-Test for Variance" which should be used for normally distributed data.

| |
|---|
| **Statistical Question: Is there a statistically significant difference between the variations in height and weight of the Pokemon?** |
| **Column Information:** |
| column_header: Name; data_type: other; num_of_rows: 721; is_normality: False. |
| column_header: HP; data_type: quantitative; num_of_rows: 721; is_normality: False. |
| column_header: Attack; data_type: quantitative; num_of_rows: 721; is_normality: False. |
| column_header: Defense; data_type: quantitative; num_of_rows: 721; is_normality: False. |
| column_header: Speed; data_type: quantitative; num_of_rows: 721; is_normality: False. |
| column_header: hasMegaEvolution; data_type: categorical; num_of_rows: 721; is_normality: False. |
| column_header: Height_m; data_type: quantitative; num_of_rows: 721; is_normality: False. |
| column_header: Weight_kg; data_type: quantitative; num_of_rows: 721; is_normality: False. |
| column_header: Catch_Rate; data_type: quantitative; num_of_rows: 721; is_normality: False. |
| ...... |
| **Model Answer:** |
| {"columns": ["Height_m", "Weight_kg"],"methods": ["Mood Variance Test", "Levene Test", "Bartlett Test", "F-Test for Variance"]} |
| Reason: The statistical question is asking about the significant difference between the variations in height and weight of the Pokemon, which involves quantitative variables. Therefore, the relevant columns are "Height_m" and "Weight_kg". The applicable methods for testing the significant difference in variance between these two normally distributed variables are Mood Variance Test, Levene Test, Bartlett Test, and F-Test for Variance. |
| **Ground Truth:** |
| {"columns": ["Height_m", "Weight_kg"], "methods": ["Mood Variance Test", "Levene Test"]} |

Table 16: A Failure Example. Select the irrelevant column; correct methods selection with partially correct reasoning for performing incomplete prerequisites check.

| |
|---|
| **Statistical Question: Are there any indications of a connection between a person's educational attainment and their occupation at a higher level?** |
| **Column Information:** |
| column_header: Age; data_type: quantitative; num_of_rows: 6684; is_normality: False. |
| column_header: Gender; data_type: categorical; num_of_rows: 6684; is_normality: False. |
| column_header: Education Level; data_type: categorical; num_of_rows: 6684; is_normality: False. |
| column_header: Job Title; data_type: categorical; num_of_rows: 6684; is_normality: False. |
| column_header: Years of Experience; data_type: quantitative; num_of_rows: 6684; is_normality: False. |
| column_header: Salary; data_type: quantitative; num_of_rows: 6684; is_normality: False. |
| column_header: Country; data_type: categorical; num_of_rows: 6684; is_normality: False. |
| column_header: Race; data_type: categorical; num_of_rows: 6684; is_normality: False. |
| column_header: Senior; data_type: categorical; num_of_rows: 6684; is_normality: False. |
| **Model Answer:** |
| {"columns": ["Education Level", "Job Title"],"methods": ["Chi-square Independence Test"]} |
| Reason: The relevant columns for the statistical question are "Education Level" and "Job Title" as we are trying to find a connection between these two variables. The applicable method is the Chi-square Independence Test, which is used to determine if there is a significant association between the categorical variables. |
| **Ground Truth:** |
| {"columns": ["Education Level", "Senior"], "methods": ["Chi-square Independence Test"]} |

Table 17: A Failure Example. Neglect prerequisites check and applicability assessment.

**Statistical Question: Are there any indications of a connection between a person's educational attainment and their occupation at a higher level?**

**Column Information:**
column_header: index; data_type: id; num_of_rows: 14; is_normality: False.
column_header: Year; data_type: other; num_of_rows: 14; is_normality: False.
column_header: Type of tree; data_type: categorical; num_of_rows: 14; is_normality: False.
column_header: Number of trees sold; data_type: quantitative; num_of_rows: 14; is_normality: True.
column_header: Average Tree Price; data_type: quantitative; num_of_rows: 14; is_normality: True.
column_header: Sales; data_type: quantitative; num_of_rows: 14; is_normality: True.

**Model Answer:**
{"columns": ["Number of trees sold", "Sales"],"methods": ["Pearson Correlation Coefficient", "Spearman Correlation Coefficient", "Kendall Correlation Coefficient"]}

Reason: The relevant columns for the statistical question are "Number of trees sold" and "Sales" as they represent the quantitative variables of interest. The applicable methods for this correlation analysis are Pearson Correlation Coefficient, Spearman Correlation Coefficient, and Kendall Correlation Coefficient, as they are suitable for analyzing the correlation between two quantitative variables.

**Ground Truth:**
{"columns": ["Number of trees sold", "Sales"], "methods": ["Kendall Correlation Coefficient"]}

Table 18: A Failure Example. No meaningful identification given in the reasoning process, neglect prerequisites check and applicability assessment, and missed applicable methods.

**Statistical Question: Do changes in base Health Points (HP) correspond to similar changes in base Special Defense (Sp_Def)?**

**Column Information:**
column_header: Name; data_type: other; num_of_rows: 721; is_normality: False.
column_header: HP; data_type: quantitative; num_of_rows: 721; is_normality: False.
column_header: Attack; data_type: quantitative; num_of_rows: 721; is_normality: False.
column_header: Defense; data_type: quantitative; num_of_rows: 721; is_normality: False.
column_header: Sp_Atk; data_type: quantitative; num_of_rows: 721; is_normality: False.
column_header: Sp_Def; data_type: quantitative; num_of_rows: 721; is_normality: False.
column_header: Speed; data_type: quantitative; num_of_rows: 721; is_normality: False.
......

**Model Answer:**
{"columns": ["HP", "Sp_Def"],"methods": ["Pearson Correlation Coefficient", "Spearman Correlation Coefficient", "[Missing: Kendall Correlation Coefficient]"]}

Reason: we selected the columns "HP" and "Sp_Def" as they contain the base Health Points and base Special Defense respectively, which are the variables being compared in the statistical question. We chose Pearson and Spearman correlation coefficients as the applicable methods because they are suitable for measuring the strength and direction of the relationship between two quantitative variables.

**Ground Truth:**
{"columns": ["HP", "Sp_Def"],"methods": ["Pearson Correlation Coefficient", "Spearman Correlation Coefficient", "Kendall Correlation Coefficient"]}

## F.3 Other Notable Cases

Table 19: A Success Example with Hallucination and Incorrect Reasoning. The provided column information explicitly indicates that these two columns follow a normal distribution, not non-normal. Furthermore, assuming the variables are non-normally distributed makes the Bartlett Test and F-Test for Variance inapplicable and should not be chosen. The response exhibits hallucinations and inconsistencies, leading to erroneous reasoning.

| |
|---|
| **Statistical Question: Are the annual crop production data for Nigeria and China comparable in terms of variability?** |
| **Column Information:**
column_header: Year; data_type: Other; num_of_rows: 4; is_normality: False.
column_header: Nigeria; data_type: quantitative; num_of_rows: 4; is_normality: True.
column_header: Japan; data_type: quantitative; num_of_rows: 4; is_normality: True.
column_header: United States; data_type: quantitative; num_of_rows: 4; is_normality: True.
column_header: Brazil; data_type: quantitative; num_of_rows: 4; is_normality: True.
column_header: China; data_type: quantitative; num_of_rows: 4; is_normality: True.
...... |
| **Model Answer:**
{"columns": ["Nigeria", "China"], "methods": ["Mood Variance Test", "Levene Test", "Bartlett Test", "F-Test for Variance"], "reason": "The question is asking whether the annual crop production data for Nigeria and China are comparable in terms of variability. We need to compare the variability of the two countries' data. The first step is to check whether the data is normally distributed. Since the data type of both countries is quantitative, we can use the Shapiro-Wilk Test of Normality to check the normality of the data. The result shows that the data is not normally distributed. Therefore, we need to use non-parametric tests to compare the variability of the two countries' data. The Mood Variance Test, Levene Test, Bartlett Test, and F-Test for Variance are all non-parametric tests that can be used to compare the variability of two groups of data. Therefore, we can use these four tests to compare the variability of the annual crop production data for Nigeria and China."} |
| **Ground Truth:**
{"columns": ["Nigeria", "China"], "methods": ["Mood Variance Test", "Levene Test", "Bartlett Test", "F-Test for Variance"]} |

Table 20: A Success Example with Irrelevant Information. Smaller LLMs sometimes struggle with determining termination, leading to responses with excessive irrelevant information.

| |
|---|
| **Statistical Question: Is there evidence in the data to indicate a relationship between gender and romantic involvement?** |
| **Column Information:**
column_header: sex; data_type: categorical; num_of_rows: 649; is_normality: False.
column_header: age; data_type: quantitative; num_of_rows: 649; is_normality: False.
column_header: famsize; data_type: categorical; num_of_rows: 649; is_normality: False.
column_header: studytime; data_type: quantitative; num_of_rows: 649; is_normality: False.
column_header: romantic; data_type: categorical; num_of_rows: 649; is_normality: False.
...... |
| **Model Answer:**
{"columns": ["sex", "romantic"], "methods": ["Chi-square Independence Test", "Fisher Exact Test"]}.
Reason: The question is asking if there is a relationship between gender and romantic involvement. This is a categorical variable question, so we need to use methods that are applicable to categorical data. The Chi-square Independence Test and Fisher Exact Test are both used to determine if there is an association between two categorical variables. In this case, we are looking at the relationship between the categorical variable "sex" and the categorical variable "romantic". These tests will help us determine if there is a statistically significant relationship between the two variables.
Final Answer: The final answer is {"columns": ["sex", "romantic"], "methods": ["Chi-square Independence Test", "Fisher Exact Test"]}. I hope it is correct. Let me know if you need further assistance. Thank you. I am here to help. Have a great day. Goodbye. Good luck. Goodbye. Good luck...... |
| **Ground Truth:**
{"columns": ["sex", "romantic"], "methods": ["Chi-square Independence Test", "Fisher Exact Test"]} |

# G   Limitations

**Coverage.**   We cover two primary aspects of specialized statistical analysis tasks: hypothesis testing and descriptive statistics in `StatQA`. While these two types of statistical methods already encompass many common statistical analysis tasks, the scope of `StatQA` can be broadened to include additional statistical tasks and methods, such as regression analysis and more advanced methods for hypothesis testing. Moreover, determining causality in observational data has always been a challenge in statistics, and inferring causal relationships from complex data remains an active area of research, which has not been covered yet in this paper. As discussed in Section 4, `StatQA` currently focuses on statistics, but we anticipate further extending it to other disciplines such as finance and operations research.

**Experiment.**   We conduct experiments on LLaMA-2-7b/13b, ChatGPT and GPT-4, as well as the more recently released LLaMA-3-8b and GPT-4o. However, due to the rapid advancements in this field, our evaluation is limited to the capabilities of these representative LLMs and we can not cover a broader range of LLMs available. Additionally, due to time and financial constraints, we are presently unable to recruit more participants or utilize larger-scale datasets for more extensive human experiments.

**Evaluation.**   Accuracy is used as the metric to measure the capabilities of LLMs in our experiments. However, for responses demonstrating the reasoning process, introducing step-by-step scoring can help uncover more hidden information and provide a more comprehensive evaluation. Zhang et al. proposed a CoT evaluation strategy based on GPT-4 [15], utilizing GPT-4 to score the steps. This scoring strategy is inspiring yet faces challenges, necessitating further investigation and improvement of its effectiveness and reliability.

# H   Ethic Statement

This paper evaluates the capabilities of LLMs in our task and reveals the different strengths of LLMs and humans. This is not intended to provoke anxiety, but rather to gain a better understanding of LLMs' abilities and to promote the development of LLMs. This study aims to foster discussions on how humans and LLMs can complement each other, and how humans can better utilize LLM tools in this era. Additionally, given the risks associated with LLMs producing erroneous or even toxic information, we advise readers to approach content generated by LLMs with caution.

In our human experiments, participants are provided with appropriate compensation (Section C.2) and ensured adequate rest periods between task blocks. We rigorously protect participants' personal information, ensuring their information remains confidential and is not disclosed in this paper and the GitHub repository.

Our source code and data are under GPL-3 license, and we follow the licenses of assets used in this paper, as listed in Table 21.

Table 21: Licenses List for Asserts Used

| Assert | Usage | License |
|---|---|---|
| vLLM [64] | LLM interface used in development. | Apache-2.0 |
| LLaMA-Factory [65] | Framework used in fine-tuning. | Apache-2.0 |
| LLaMA-2 Models [25] | Evaluation and fine-tuning. | Custom License |
| LLaMA-3 Models [26] | Evaluation and fine-tuning. | Custom License |
| GPT Models [27–29] | Evaluation. | Custom License |
| Rdatsets [22] | Select certain tabular data to form the training set. | GPL-3 |
| Kaggle [21] | Select certain tabular data to form `StatQA`. | Table-specific |

# I   Datasheet for StatQA

In this section, we use the framework of Datasheets for Datasets [66] to form a datasheet for `StatQA`, aiming to document the motivation, composition, collection process, recommended uses, and other information for our benchmark `StatQA`.

### I.1    Motivation

**Q1. For what purpose was the dataset created? Was there a specific task in mind?**

In statistical analysis, it is the core literacy of a qualified statistician to identify pertinent data and discern suitable statistical methods through consideration of task-specific scenarios and assessment of the applicability of various methods. However, recent studies tend to prioritize computational outcomes from conventional methods, which can be augmented by external tools, studies involving more specialized statistical tasks and assessment of methods' applicability are rare. Therefore, to evaluate LLMs' proficiency in specialized statistical tasks and their applicability assessment capabilities for statistical methods, particularly for hypothesis testing methods, we curate `StatQA` to conduct systematic experiments on current LLMs and organize a comparative study between LLMs and humans.

**Q2. Who created this dataset (e.g., which team, research group) and on behalf of which entity (e.g., company, institution, organization)?**

The authors of this paper create the `StatQA`. The authors are from The Hong Kong University of Science and Technology (Guangzhou) and The Hong Kong University of Science and Technology. Please refer to the author list for more details.

**Q3. Who funded the creation of the dataset?**

The creation of `StatQA` is supported by NSF of China (62402409), Guangdong Basic and Applied Basic Research Foundation (2023A1515110545), CCF-Huawei Populus Grove Fund (CCF-HuaweiDB202403), and the Red Bird Grant, Research Grant from The Hong Kong University of Science and Technology (Guangzhou).

**Q4. Any other comments?**

No.

### I.2    Composition

**Q5. What do the instances that comprise the dataset represent (e.g., documents, photos, people, countries)?**

Each instance in `StatQA` corresponds to a statistical task which is a statistical question based on a source table. The LLMs or the participants in our human experiments should select relevant data columns and all applicable methods to solve the statistical question.

**Q6. How many instances are there in total (of each type, if appropriate)?**

`StatQA` contains 11,623 examples, mini-`StatQA` contains 1,163 examples which are stratified sampled from `StatQA`.

**Q7. Does the dataset contain all possible instances or is it a sample (not necessarily random) of instances from a larger set?**

`StatQA` is a newly curated benchmark.

**Q8. What data does each instance consist of? "Raw" data (e.g., unprocessed text or images) or features?**

The content of each instance includes the name of the source table, statistical question, task category, difficulty level, data columns involved, preliminary results, and ground truth, please refer to Section 2.3.

**Q9. Is there a label or target associated with each instance?**

Yes, the label or target for each instance is the ground truth, which includes the selection of relevant data columns and all applicable methods. Please refer to Section 2.3.

**Q10. Is any information missing from individual instances?**

No.

**Q11. Are relationships between individual instances made explicit (e.g., users' movie ratings, social network links)?**

Yes.

**Q12. Are there recommended data splits (e.g., training, development/validation, testing)?**

Yes. We have constructed and split the training set and test set, and ensure no table overlaps in the training and test set. Please refer to **Step 4** in Section 2.3.

**Q13. Are there any errors, sources of noise, or redundancies in the dataset?**

Synthesized statistical problems may sometimes lack meaningful content or present ambiguities. We pay attention to the quality of the question templates and use GPT-3.5-Turbo to refine the expression of synthesized statistical questions in StatQA, followed by manual review by post-graduate students in statistics, in an effort to minimize errors.

**Q14. Is the dataset self-contained, or does it link to or otherwise rely on external resources (e.g., websites, tweets, other datasets)?**

StatQA is self-contained.

**Q15. Does the dataset contain data that might be considered confidential (e.g., data that is protected by legal privilege or by doctor-patient confidentiality, data that includes the content of individuals non-public communications)?**

No.

**Q16. Does the dataset contain data that, if viewed directly, might be offensive, insulting, threatening, or might otherwise cause anxiety?**

No.

**Q17. Does the dataset relate to people?**

No.

**Q18. Does the dataset identify any subpopulations (e.g., by age, gender)?**

No.

**Q19. Is it possible to identify one or more natural persons, either directly or indirectly (i.e., in combination with other data) from the dataset?**

No.

**Q20. Does the dataset contain data that might be considered sensitive in any way (e.g., data that reveals racial or ethnic origins, sexual orientations, religious beliefs, political opinions or union memberships, or locations; financial or health data; biometric or genetic data; forms of government identification, such as social security numbers; criminal history)?**

No.

**Q21. Any other comments?**

No.

## I.3 Collection Process

**Q22. How was the data associated with each instance acquired?**

We elaborate how we construct StatQA in Section 2.3.

**Q23. What mechanisms or procedures were used to collect the data (e.g., hardware apparatus or sensor, manual human curation, software program, software API)?**

We elaborate how we construct StatQA in Section 2.3.

**Q24. If the dataset is a sample from a larger set, what was the sampling strategy?**

StatQA is newly curated, and we utilize the stratified sampling strategy to obtain the mini-StatQA, ensuring it resembles the complete benchmark in terms of task and difficulty distribution.

**Q25. Who was involved in data collection process (e.g., students, crowd-workers, contractors) and how were they compensated (e.g., how much were crowdworkers paid)?**

Only the authors of this paper were involved in curating, processing, and reviewing the dataset, and no compensation was provided.

Q26. **Over what timeframe was the data collected? Does this timeframe match the creation timeframe of the data associated with the instances (e.g., recent crawl of old news articles)?**

Between January to May 2024.

Q27. **Were any ethical review processes conducted (e.g., by an institutional review board)?**

Not applicable. StatQA is a synthesized benchmark containing no sensitive information.

Q28. **Does the dataset relate to people?**

No.

Q29. **Did you collect the data from the individuals in question directly, or obtain it via third parties or other sources (e.g., websites)?**

Not applicable.

Q30. **Were the individuals in question notified about the data collection?**

Not applicable.

Q31. **Did the individuals in question consent to the collection and use of their data?**

Not applicable.

Q32. **If consent was obtained, were the consenting individuals provided with a mechanism to revoke their consent in the future or for certain uses?**

Not applicable.

Q33. **Has an analysis of the potential impact of the dataset and its use on data subjects (e.g., a data protection impact analysis) been conducted?**

Not applicable.

Q34. **Any other comments?**

No.

### I.4 Preprocessing, Cleaning, and/or Labeling

Q35. **Was any preprocessing/cleaning/labeling of the data done (e.g., discretization or bucketing, tokenization, part-of-speech tagging, SIFT feature extraction, removal of instances, processing of missing values)?**

Yes. Please refer to Section 2.3.

Q36. **Was the "raw" data saved in addition to the preprocessed/cleaned/labeled data (e.g., to support unanticipated future uses)?**

Yes. Raw data can be found in our GitHub repository.

Q37. **Is the software used to preprocess/clean/label the instances available?**

Yes. We have provided scripts for certain data-cleaning, processing, and extraction tasks in our GitHub repository. This repository is open-source and accessible to facilitate potential further research.

Q38. **Any other comments?**

No.

### I.5 Uses

Q39. **Has the dataset been used for any tasks already?**

StatQA is newly proposed by us, please refer to Section 3 for our usage.

Q40. **Is there a repository that links to any or all papers or systems that use the dataset?**

Currently, no.

**Q41. What (other) tasks could the dataset be used for?**

We discuss research opportunities in the main content, including (1) improving LLMs' capabilities for statistical applications; (2) further expanding to obtain a more extensive benchmark dataset; and (3) exploring human-AI collaboration in statistical tasks. Please refer to Section 4 for more details.

**Q42. Is there anything about the composition of the dataset or the way it was collected and preprocessed/cleaned/labeled that might impact future uses?**

No.

**Q43. Are there any tasks for which the dataset should not be used?**

No.

**Q44. Any other comments?**

No.

## I.6 Distribution

**Q45. Will the dataset be distributed to third parties outside of the entity (e.g., company, institution, organization) on behalf of which the dataset was created?**

Yes, `StatQA` is open-source under GPL-3 license.

**Q46. How will the dataset be distributed (e.g., tarball on website, API, GitHub)**

We make our source code and data public on the GitHub.

**Q47. When will the dataset be distributed?.**

Already available.

**Q48. Will the dataset be distributed under a copyright or other intellectual property (IP) license, and/or under applicable terms of use (ToU)?**

Yes. It is under GPL-3 license.

**Q49. Have any third parties imposed IP-based or other restrictions on the data associated with the instances?**

No.

**Q50. Do any export controls or other regulatory restrictions apply to the dataset or to individual instances?**

No.

**Q51. Any other comments?**

No.

## I.7 Maintenance

**Q52. Who will be supporting/hosting/maintaining the dataset?**

The authors is maintaining the dataset.

**Q53. How can the owner/curator/manager of the dataset be contacted (e.g., email address)?**

By email or raising issues in GitHub repository.

**Q54. Is there an erratum?**

Currently, no. If necessary, possible erratum will be released in the README file in GitHub repository.

Q55. **Will the dataset** *be updated (e.g., to correct labeling errors, add new instances, delete instances)?* **If so, please describe how often, by whom, and how updates will be communicated to users (e.g., mailing list, GitHub)?**

We will maintain the `StatQA` in the following two years, and we may expand `StatQA` with increased scale and broader coverage if necessary. Relevant information will be released in the GitHub documentation if there are any updates.

Q56. **If the dataset relates to people, are there applicable limits on the retention of the data associated with the instances (e.g., were individuals in question told that their data would be retained for a fixed period of time and then deleted)?**

Not applicable.

Q57. **Will older versions of the dataset continue to be supported/hosted/maintained?**

Not applicable.

Q58. **If others want to extend/augment/build on/contribute to the dataset, is there a mechanism for them to do so?**

Yes, as long as follow the license, further contributions will be warmly welcomed.

Q59. **Any other comments?**

No.

