# OpenReview forum: "Are Large Language Models Good Statisticians?"
_NeurIPS.cc/2024/Datasets_and_Benchmarks_Track — NeurIPS 2024 Track Datasets and Benchmarks Poster_

### Official Review · Reviewer_8PyK · 2024-07-05
**Assessing the effectiveness of LLMs for applying statistical methods**

**Rating:** 7
**Confidence:** 3

**Review:**

The study presented in the paper for LLMs in statistical analysis tasks is interesting and provide new insights into the effectiveness of current SoTA LLMs in performing statistical analysis tasks, especially when it comes to the applicability of different methods. A novel benchmark is also presented that could promote more research on this topic in the future.

I do see some issues with the experiments and the overall language in the paper (spelling errors, sentence structure, etc.) that should be addressed, as described in the rest of the review.

**Strengths:**

The paper presents a novel study in the use of LLMs for statistical analysis tasks, which may have impact on the LLM community in general. I also think that the paper would be the go-to citation for anyone who would want to discuss the effectiveness of LLMs in applied statistics, since I did not find much existing literature on the subject. The benchmark may also be useful to anyone who wants to conduct research into using LLMs for hypothesis testing.

**Additional Feedback:**

Currently, only GPT and Llama models are considered in the experiments. While these are SoTA methods, it may be useful to consider some more models, and even some smaller models for baselines as well (for many who may be interested in fine-tuning approaches, GPT/Llama may not be viable options due to the required computational resources).

I think this paper should be accepted into NeurIPS, but I hope the authors will further improve the paper and take my feedback into account.

**Clarity:**

I think the paper is well written in overall, however there are some spelling errors such as:

*Line 10 in the abstract: "achieve a best performance of only 64.83%". I think the corresponding statistic where the score was reached should be mentioned.

*Figure 2 "Results winclude applicable statistical methods [...]".

*Line 264: "LLMs are good at distinguishing different statistical tasks then selecting associated
methods but struggle to utilize the domain knowledge to assess method applicability effectively"



In Table 2's caption, reminding the reader what CA, CTT, DCT etc. stand for could be useful.

**Correctness:**

I did not find any clear (symbolic) definition of how accuracy is calculated in the experiments. Accuracy is mentioned in 173-176, but not really explicitly defined in terms of symbols.

I also think that there should be error bars in Table 2, so that it is possible to determine whether or not differences are significant.

It may also be relevant to measure precision and recall in addition to accuracy to catch any imbalances in the data.

In overall, I still believe that the claims made in the submission are correct.

**Documentation:**

The appendix contains more details regarding related work, more details regarding the experiments (such as LLM prompts), and documentation of the benchmark. The benchmark is available via github, which also has detailed instructions for setup and installation.

I think the benchmark and the submission in overall is well documented.

**Ethics:**

I don't see any immidiate negative impacts from anything presented in this paper. The authors also have ethics and limitation statements in the appendix, which I think describe these matters sufficiently.

**Limitations:**

I think there could be one or two examples/analysis in the paper of how LLM's perform applicability errors. I see that this is included in the appendix, but I think it might be useful to discuss it in the main paper.

I think the benchmark could be even more relevant if it also deals with causality, such as discovery or reasoning. See the QRData benchmark.

**Opportunities For Improvement:**

The link to the benchmark can be found at line 65, but I don't think it is easy enough to locate. Please consider making it more visible, such as placing it in a footnote at page 1.

Also, see my comments below.

**Relation To Prior Work:**

To me, the relationship with previous work and contributions seems clear. The authors have not included a related work-section in the paper, but done so in the appendix. I think this is appropriate and a good decision.

I did not find much existing literature on the subject. There are previous work that look into causality and relations (also mentioned by the authors), but these don't deal with statistics. There are also two papers that look into LLMs for statistics (mentioned by the authors in the appendix), but they do not consider the applicability aspect of statistical methods.

Thus, i think that the paper addresses a gap in the current SoTA and would likely be useful as a citation source for other researchers.

**Summary And Contributions:**

The paper presents a benchmark for statistical analysis tasks called StatQA, for performing evaluations of LLM models in statistical hypotheses testing. Evaluations of LLMs show that there is room for improvement in the current SoTA models in terms of carrying out hypothesis testing and analysis. Experiments where LLMs are compared to human performance show that the types of errors that are made differ between the two groups.

---

> ### Author Rebuttal · Authors · 2024-08-15
>
> Dear Reviewer,
>
> Thank you very much for your time and constructive suggestions, we are pleased and encouraged by your recognition. Below, we will try our best to address the issues raised, hoping our responses can help to clarify.
>
> **1. Clarity part**
>
> - We would like to express our sincere gratitude for your help in pointing out typos and expression problems! We will improve these expressions based on your feedback, make our repository links easier to find, and add explanations to abbreviations in Table 2 for better clarity. In addition, we conducted a thorough check of our manuscript to correct other potential problems.
> - Due to the length limitation at the submission stage, we have to temporarily place error examples in the appendix. If an additional page can be obtained, we will be pleased to present it in the main part, which we believe will be clearer.
>
> **2. Correctness part**
>
> - In Section 3.1, we mentioned that our metric Acc(C, M) refers to the proportion of methods and column selections fully aligned with the ground truth without any omissions or incorrect selections. In our setup, actually our Acc(C,M) metric is more analogous to the *Exact Match (EM) metric*, which to measure the proportion of cases where the predicted output fully matches the ground truth.
>
>     For Acc(C,M), it is calculated by cases with fully correct columns and methods selection (both fully match with ground truth) divided by the total number of cases. Hereby, its formal definition in terms of symbols is the following:
>
>     $Acc(\Bbb C,\Bbb M)=\frac{\sum_{i=1}^N 1(\hat{\Bbb C_i}=\Bbb C_i, \hat{\Bbb M_i}=\Bbb M_i)}{N}$, where N means the total number of test examples.
>
>     Therefore, given the specificity of our metric, it may not be the most applicable way to apply recall and precision to our task. We understand your concern about potential imbalances. In our early practice, we also found that the balance of the dataset is important, which affects the the discriminative ability of our benchmark. To address it, we introduced the balancing operation in Step 4 of dataset construction, by controlling different sampling parameters, expanding the underrepresented types while sampling the abundant ones, to make our dataset balanced in terms of example types and difficulty distributions.
>
> - Considering a complete round of experiments requires a considerable overhead of computing resources, we minimize the randomness as much as possible by setting the hyperparameter temperature to 0 to obtain more stable results, as we mentioned in appendix Section C.
>
> **3. Performance of smaller open-source LLMs**
>
> Thanks for this constructive advice! We fully understand that many potential users may only have very limited resources, so if smaller models can achieve decent performance, it can be helpful for them. Therefore, we conducted evaluations on much smaller open-source LLMs, which are qwen2-0.5b-instruct and qwen2-1.5b-instruct. We demonstrate the 0-shot and 1-shot overall Acc(C,M) of these two representative models as follows:
>
> | model | 0-shot | 1-shot |
> | --- | --- | --- |
> | qwen2-0.5b-inst | 1.38% | 2.41% |
> | qwen2-1.5b-inst | 15.99% | 20.38% |
>
> The results indicate that the qwen2-0.5b-instruct is incapable of our task. However, qwen2-1.5b-instruct shows comparable abilities to some LLaMA models, with performance between LLaMA-2-13b and LLaMA-3-8b. Considering qwen2-1.5b-instruct is much smaller in parameters, although its performance is far from “strong”, we still think it is an exciting and encouraging result for users with very limited computational resources.
>
> **4. Related work**
>
> > I think the benchmark could be even more relevant if it also deals with causality, such as discovery or reasoning. See the QRData benchmark.
> >
>
> Thanks for this advice and for highlighting the relevance of causality. We also found this work inspiring and discussed QRData in related work (Section 5 and appendix A.2). As we discussed in limitations (appendix Section F), we leave this expansion of casual reasoning examples to future work.
>
> ---
>
> As per latest NeurIPS rules, we are unable to upload a revised version during the rebuttal period, but we will certainly improve our manuscript based on your invaluable suggestions. Also, we look forward to continuously improving our dataset and we will release an enhanced version 2.0 in the near future.
>
> If something is still unclear, welcome to continue the discussion with us. Once again, thank you very much for your insightful feedback!

---

> > ### Comment · Reviewer_8PyK · 2024-08-20
> >
> > I think the authors provide a good feedback on the review. I was unaware that it is not possible to revise the paper during the rebuttal period for this year's NeurIPS. Thus, I will stay on my current review/score which is a clear recommendation for accepting the paper into NeurIPS.

---

### Official Review · Reviewer_jPG1 · 2024-07-24
**Test LLMs for statistical analyses.**

**Rating:** 6
**Confidence:** 4
**Correctness:** Check “Opportunities for improvement”
**Clarity:** Check “Opportunities for improvement”

**Review:**

This work proposes StatQA aiming to fill the gap of testing the capabilities of LLMs in conducting statistical analyses, especially, hypothesis test methods. It provide more comprehensive tests than existing one. The experimental results are interesting and inspiring. More clarifications are needed.

**Strengths:**

* A new benchmark for statistical analysis tasks, with a particular focus on assessing the applicability of hypothesis testing methods.

* Evaluate widely used LLMs to establish benchmarks for the statistical tasks in StatQA. Domain-specific prompts and fine-tuning were tried to improve LLMs for these tasks.

* Compare the performance of Humans and LLMs

**Additional Feedback:**

None

**Documentation:**

Yes

**Limitations:**

More discussions about limitations are needed.

**Opportunities For Improvement:**

* Elaborate how "easy" and "hard" were decided? Are they defined for human or LLMs?

* Why only use mini-StatQA in experiments.

* Why refined Questions by GPT-3.5-Turbo? Will this create bias (e.g., making some tasks easier for human/LLMs)?

* In “Key Ideas for Developing StatQA”, does “synthesize statistical questions” mean that the questions were automatically generated according to the designed templates?

* How were error analyzed? By human experts? Is there a protocol?

**Relation To Prior Work:**

Yes

**Summary And Contributions:**

LLMs excel in many scientific fields but have not been systematically tested with complex statistical tasks. The StatQA benchmark was created, featuring 11,623 examples, to test LLMs' proficiency in specialized statistical tasks and applicability assessment capabilities, especially for hypothesis testing methods. Tests show that even top models like GPT-4o achieve only 64.83%, highlighting significant improvement potential. Fine-tuned LLMs outperform standard ones, and error analysis reveals LLMs often make applicability errors, while humans confuse statistical tasks. This suggests combining LLM and human expertise could be beneficial, meriting further investigation.

---

> ### Author Rebuttal · Authors · 2024-08-15
>
> Dear Reviewer,
>
> Thank you very much for your time and constructive suggestions. We are pleased and encouraged that our research motivation, dataset, evaluation, and analysis are recognized. Below, we will try our best to address the issues raised, hoping our responses can help to clarify.
>
> **1. Label for difficulty level**
>
> Thanks for pointing out! Difficulty labels are automatically labeled in data synthesis according to the property of the question. Generally speaking, if a question requires additional and accurate applicability assessment to avoid mistakes, it will be labeled as hard; otherwise, it will be labeled as easy. We hope the following example can further explain and make it clearer:
>
> Assuming we have two questions of variance test, Q1 (involves two columns of C1 and C2) and Q2 (involves C3 and C4). Suppose C1 and C2 are normally distributed while C3 and C4 are not. In this case, all our listed variance methods M1, M2, M3, and M4 are applicable to Q1, but since methods M3 and M4 should be used on normally-distributed data, only M1 and M2 are applicable to Q2. In summary, for Q1: M1, M2, M3 and M4 are all applicable; for Q2: M1 and M2 are applicable but M3 and M4 are not. Therefore, we label question Q2 as “hard” for additional challenges are placed on the LLM to assess the applicability, and LLMs are likely to select inappropriate methods if they fail to correctly assess. In contrast, Q1 is not such challenging, so we label it as “easy”.
>
> **2. Why use mini-StatQA in evaluation?**
>
> For a more efficient evaluation and to help users with limited resources, we provide mini-StatQA with 1,163 examples, which are stratified sampled from StatQA, ensuring they are distributionally proximate. We believe mini-StatQA with more than 1,000 test examples is sufficient to test LLMs’ capabilities on our task if refer to other renowned science or math reasoning benchmarks[1][2]. In addition, as mentioned in Table 1, we also offer the StatQA of 11,623 test examples in our repository for users with sufficient computational resources.
>
> [1] Mathematical Capabilities of ChatGPT
>
> [2]SciBench: Evaluating College-Level Scientific Problem-Solving Abilities of Large Language Models
>
> **3. Why GPT-3.5-Turbo as the refiner?**
>
> As presented in Section 2.3, we introduce GPT-3.5-Turbo to refine and rephrase statistical questions without changing original meanings. In detail, the goals of this task are to 1) correct potential English grammar mistakes (e.g. singular and plural forms); 2) make the sentence semantically coherent (e.g. mitigating the possible expression rigidity and artificiality when using templates); 3) increase the diversity of expressions.
>
> This task is more of a requirement for English proficiency and text processing instead of statistical knowledge. In practice, we examined the refined results, and we found although GPT-3.5-Turbo was not as strong as GPT-4 in statistical tasks, it still showed satisfactory English proficiency and has been very capable of handling relatively simple text processing tasks like sentence rephrasing here to meet our goals above.
>
> Additionally, as mentioned in task goals, the refinement of original questions focuses on rephrasing expressions while maintain the original question intent, we have not detected potential bias in manual checking.
>
> **4. About question synthesis**
>
> > In “Key Ideas for Developing StatQA”, does “synthesize statistical questions” mean that the questions were automatically generated according to the designed templates?
> >
>
> Yes, your understanding is generally accurate. We bring the involved columns’ headers into the question templates (we prepared a variety of question templates for each scenario to randomly choose, more than 10 for each scenario) to replace the placeholder in templates to generate original questions and subsequently refine them to obtain more coherent and diverse expressions. We demonstrate some examples (not all templates) of question templates in Table 4 in the appendix Section B.3.
>
> **5. About error analysis**
>
> We designed automated programs to identify the four categories of errors mentioned in Section 3.3: (1) Invalid Answer; (2) Column Selection Error; (3) Statistical Task Confusion; (4) Applicability Error; and mixed errors.
>
> For example, assuming we have a question Q of the contingency table test task, the correct relevant columns are C1 and C2, and its applicable method is “Fisher Exact Test”. If the LLM selects columns other than C1 and C2, it would be Column Selection Error; if the LLM selects methods not belong to the contingency table test, like “Pearson Correlation Coefficient” which is correlation analysis, it would be Statistical Task Confusion; if the LLM selects methods in the contingency table test, but inapplicable ones, then it would be Applicability Error.
>
> The identification and statistics of errors will be conducted by our designed program. We provide more examples and explanations for different types of errors in the appendix, please refer to Table 9 in Section D, hoping it can provide more details.
>
> **6. More discussions about limitations**
>
> Thank you for this advice! We discussed limitations in the appendix Section F, including three aspects of limitations: 1) dataset coverage; 2) extent of LLMs and human experiments; and 3) the granularity of evaluation. We also plan to add discussions on fairness and the issue that LLMs can lead to errors or information is not aligned with human values, and we will continuously improve this part.
>
> ---
>
> As per latest NeurIPS rules, we are unable to upload a revised version during the rebuttal period, but we will certainly improve our manuscript based on your invaluable suggestions. Also, we look forward to continuously improving our dataset and we will release an enhanced version 2.0 in the near future.
>
> If something is still unclear, welcome to continue the discussion with us. Once again, thank you very much for your insightful feedback!

---

> > ### Comment · Reviewer_jPG1 · 2024-08-16
> > **Difficulty level**
> >
> > Please elaborate more about how to automatically decide if "a question requires additional and accurate applicability assessment"

---

> > ### Author Rebuttal · Authors · 2024-08-17
> >
> > Thanks for your reply! We would be pleased to provide further elaboration.
> >
> > In previous response, we use an example to explain how the difficulty label are determined in our data synthesis.
> >
> > > Assuming we have two questions of variance test, Q1 (involves two columns of C1 and C2) and Q2 (involves C3 and C4). Suppose C1 and C2 are normally distributed while C3 and C4 are not. In this case, all our listed variance methods M1, M2, M3, and M4 are applicable to Q1, but since methods M3 and M4 should be used on normally-distributed data, only M1 and M2 are applicable to Q2. In summary, for Q1: M1, M2, M3 and M4 are all applicable; for Q2: M1 and M2 are applicable but M3 and M4 are not. Therefore, we label question Q2 as “hard” for additional challenges are placed on the LLM to assess the applicability, and LLMs are likely to select inappropriate methods if they fail to correctly assess. In contrast, Q1 is not such challenging, so we label it as “easy”.
> > >
> >
> > Following the labeling rules explained in our previous response, actually the difficulty labels are automatically assigned based on the applicable methods in the ground truth during dataset synthesis.
> >
> > For instance, in the case above, the variance testing task provides four candidate methods, $S_{VT}=(M1, M2, M3, M4)$, of which M1 and M2 are applicable to Q2, while M3 and M4 are not. Therefore, the ground truth for applicable methods for Q2 is $S_2 = (M1, M2) \subset S_{VT}$ (*the set of applicable methods is a proper subset of the candidate set*), and we label as "hard", as it indicates that there are unsuitable methods *eliminated* in the applicability assessment. In contrast, the ground truth for applicable methods for Q1 is $S_1 = (M1, M2, M3, M4) = S_{VT}$, so it is labeled as "easy". The process is automatically processed based on the applicable methods in the ground truth of each question in our dataset. Notably, we made some uniform adjustments for certain types of questions. For example, descriptive statistics questions (e.g. questions about mean, average, quartile, etc.) are very straightforward and rarely involve applicability assessments, so we uniformly labeled their difficulty as easy.
> >
> > We hope our elaboration has been helpful to clarify. Should you have any further questions or require additional clarification, we would be glad to discuss them further. Thanks for your reply!

---

> > > ### Comment · Reviewer_jPG1 · 2024-08-17
> > > **Difficulty level (cont)**
> > >
> > > How big is the candidate method pool? Is it possible to have random methods in S_VT of Q, which are trivial to tell that them are not applicable to Q?

---

> > ### Author Rebuttal · Authors · 2024-08-18
> >
> > Thank you very much for your follow-up questions!
> >
> > The candidate methods used to help to determine difficulty labels only includes methods with similar usage scenarios within the same task type, rather than random methods that can be trivially discerned. This makes questions with $S_{applicable} \subset S_{candidate}$ more challenging.
> >
> > For instance, in the case of variance tests mentioned above, candidate methods include only those related to variance tests. Specifically, as presented in Figure 3 and Appendix Table 3, candidate set for variance test (VT) $S_{VT} = $$($"F-Test for Variance", "Bartlett Test", "Mood Variance Test", "Levene Test"$)$, which are referred as M1-M4 in previous response for brevity. It *will not* include other random or irrelevant methods with distinctly different usage scenarios (e.g., mean, quartile, or some correlation coefficients, etc.).
> >
> > Therefore, the focus of the candidate methods lie in similarity rather than scale, with the candidate methods varying across different scenarios. Generally speaking, we classify the statistical task of the current problem and set all methods within the same category as candidates, for example there are four candidate methods for variance test questions, please refer to Figure 3 in the main text or Table 3 in the appendix. Specifically, as mentioned in our previous response, tasks that are overly straightforward and involve minimal applicability assessment (such as descriptive statistics, checks for gamma/uniform/exponential distributions) are excluded and just directly labeled as “easy”.
> >
> > Consequently, “hard” labeled questions require more in-depth understanding of statistical methods and accurate applicability assessment from LLMs to select proper methods for given questions, beyond simply classifying question category and subsequently select candidate methods indiscriminately.
> >
> > We hope our response helps to address raised concerns. Thank you for your continued engagement!

---

### Official Review · Reviewer_f7Y3 · 2024-07-24
**A benchmark for evaluating method selection capabilities of LLMs in performing statistical analysis**

**Rating:** 7
**Confidence:** 3

**Review:**

The motivation for a dataset of evaluating statistical method selection in complex statistical analysis is well justified. The dataset construction covers a wide range of methods and domains and the results reveal interesting insights on the equivalence and complementarity of the abilities of LLMs and human statisticians.

Pros:
* The dataset is diverse, with accounts for good coverage and difficulty levels, as well as relevance for the task of evaluating proficiency in the field of statistics.
* The variability of both models and human performance on individual tasks reveal potentially interesting properties, as well as highlight the importance of evaluating statistical literacy on a more granular level.
* The comparison of LLMs with human performance provides insights on the sub-tasks where complementarity between the two is recommended, as well as insights on how the current models can be used by experts more efficiently.

Cons:
* There are some sections that would require additional details, such as the "domain knowledge strategy", dataset construction pipeline and train-test split with regard source domains.

**Strengths:**

* The authors identify method selection as an overlooked stage in evaluating the abilities of LLMs to perform complex statistics and propose StatQA as a candidate.
* The dataset is diverse, covering a diverse range of statistical tasks, namely Correlation Analysis, Contingency Table Tests, Distribution Compliance Tests and Variance Tests and a diverse set of tables with column header annotations from various domains.
* The performance of LLMs is compared with human statisticians, in both closed-book and open-book setups, revealing areas of similarity and dissimilarity.

**Additional Feedback:**

How are the domains balanced in train and test out of the 78 initial tables, out of the previously mentioned domains of such datasets, including education, medicine, engineering and economy? Are all domains present in StatsQA also present in the train data? What are the ratios?

**Clarity:**

Yes, the paper is well written, with the mention that some the sections would benefit from additional details (mentioned in Opportunities For Improvement).

**Correctness:**

The claims are correct and the dataset construction is sound. Source code is provided.

**Documentation:**

Yes, there is enough detail and the code is openly available as a GitHub repository.

**Ethics:**

There are no ethical concerns with the submission.

**Limitations:**

There are mentions of a discussion on limitations in the Appendix.

**Opportunities For Improvement:**

The dataset creation pipeline is not perfectly clear and I kindly ask the authors to clarify these aspects:
 1. How are columns selected for a given statistical test M in Stage 1? Is a statistical test sampled first, then all combinations of columns are generated, while filtering out the subsets that don't meet the prerequisites of the test? Is there a faster method that is being used?
2. Why was the "hard" tag defined as whether the prerequisites of the methods are met? My understanding of the figure and explanations in section **2.3** is that question templates in step 3 are chosen based on the data columns identified in step 2, which ensures prerequisites checks.

What are the details of the "domain knowledge" strategy? How are models endowed with it, as it is referred to as a "new strategy" (line 185)? There are not enough details on this step. Additionally, domain knowledge seems to decrease some of the scores, especially on CA, for the majority of models, what would be the intuition here? Similar for CoT prompting resulting in performance degradation.

**Relation To Prior Work:**

Yes, the study provides literature review on relevant benchmarks and discusses focus on method selection and a broader coverage as a differentiating factor from previous work.

**Summary And Contributions:**

The authors propose StatQA, a benchmark for evaluating the effectiveness of large language models in conducting statistical analysis, a field that is still under-explored while larger and more capable models are developed at a fast pace. Specifically, the benchmark aims to evaluate the ability of models to select relevant data and assess the applicability of statistical testing methods given a statistical query.

The authors conduct an extensive evaluation that reveals a significant deficit of even state-of-the-art methods in handling statistical tasks of such complexity, as well as major improvements of open-source models obtained through domain specific fine-tuning. The authors further discuss potential research directions in the field and suggest areas of potential complementarity between LLMs and human experts following a detailed comparison of human performance and LLM performance on the proposed benchmark.

---

> ### Author Rebuttal · Authors · 2024-08-15
>
> Dear Reviewer,
>
> Thank you very much for your time and constructive suggestions. We are pleased and encouraged that our motivation, dataset, and analysis are recognized. Below, we will try our best to address the issues raised, hoping our responses can help to clarify.
>
> **1. Column selection**
>
> Yes, your understanding is generally accurate. Actually, we first filter out the columns that do not satisfy the prerequisites, then combine columns that meet the prerequisites for subsequent synthesis. It would be more efficient than combining all the columns first and then filtering them.
>
> **2. Label for difficulty level**
>
> We are pleased to clarify. Generally speaking, if a question requires additional and accurate applicability assessment to avoid mistakes, it will be labeled as hard; otherwise, it will be labeled as easy.
>
> Let us use an example to make it clearer: assuming we have two questions of variance test, Q1 (involves two columns of C1 and C2) and Q2 (involves C3 and C4). Suppose C1 and C2 are normally distributed while C3 and C4 are not. In this case, all our listed variance methods M1, M2, M3, and M4 are applicable to Q1, but since methods M3 and M4 should be used on normally-distributed data, only M1 and M2 are applicable to Q2. In summary, for Q1: M1, M2, M3 and M4 are all applicable; for Q2: M1 and M2 are applicable but M3 and M4 are not. Therefore, we label question Q2 as “hard” for additional challenges are placed on the LLM to assess the applicability, and LLMs are likely to select inappropriate methods if they fail to correctly assess. In contrast, Q1 is not such challenging, so we label it as “easy”.
>
> **3. Domain knowledge in prompt**
>
> - **About domain knowledge**
>
>     Introduced domain knowledge (DK) in the prompt refers to brief, hand-collated information about the usage scenarios and applicability of statistical methods, such as "F-Test for Variance: Whether there is a significant difference in variance between normally distributed variables", as shown in Table 8 in the appendix. Our reason for introducing it was to examine the performance of LLMs when this information is explicitly provided and to contrast it with when it is not. We found that domain knowledge helped to improve the capabilities, but LLMs still struggle to utilize this knowledge well to achieve satisfactory performance, indicating significant room for improvement.
>
> - **Affect of introducing domain knowledge in prompt**
>
>     Thank you for your attention to this! The inclusion of DK improve overall performance, but lower some models’ accuracy in correlation analysis (CA) and descriptive statistics (DS) task. We conducted additional error analysis to compare 1-shot and 1-shot+DK on CA task examples (*please find the chart in attached PDF*), and we noticed some of LLMs’ column selection and statistical task confusion errors slightly increased after introducing DK.
>
>     We have also analyzed affects of CoT in appendix Section D.2, uncovering CoT slightly increase the incidence of invalid answers and column selection errors on smaller LLMs like LLaMA but not prominent affect on larger LLMs like GPT. Similarly to DK, among five subtasks, CoT has more negative impact on simple DS tasks.
>
>     However, it is quite notable that DK and CoT bring a positive impact to DCT, CTT, and VT subtasks which are more complex and less commonly seen compared to CA and DS. Particularly, DK significantly boost LLMs' performance on these three subtasks, and its introduction is beneficial in terms of overall performance.
>
> - **Potential reason**
>
>     We think these phenomenon above may have the following potential reasons:
>
>     - Step-by-step thinking of CoT can be beneficial in more complex subtasks which require stronger reasoning to get the right answer;
>     - Since DCT, CTT, and VT tasks are less commonly seen compared to CA and DS, LLMs may be less involved in their pretraining data, making them relatively unfamiliar with these tasks, so DK make significant improvement;
>     - For relatively simple and straightforward tasks (CA and DS here), with which LLMs have some familiarity, the introduction of DK and CoT may cause some of the LLMs to obfuscate the information or overthink, leading to varying degrees of performance degradation on some subtasks.
>
>     Deeper reason may stem from various process of LLMs’ pre-training and the mechanics of inference. Moreover, we truly appreciate your insight into this, which is very enlightening.
>
>
> **4. About fields of source tables**
>
> We categorize the fields of the source data table into education, medicine, science, engineering, economy, and life. Our goal is to ensure that we cover all domains when splitting the training and test sets based on these source tables. For source tables used for testing (to form StatQA for LLM evaluation) and for training (to form D_train for fine-tuning), the numbers of tables in different fields are as follows:
>
> | Field/#-tables | StatQA (for test) | D_train (for training) |
> | --- | --- | --- |
> | Education | 9 | 5 |
> | Medicine | 4 | 7 |
> | Science | 4 | 3 |
> | Engineering | 3 | 4 |
> | Economy | 10 | 13 |
> | Life | 7 | 9 |
>
> ---
>
> As per latest NeurIPS rules, we are unable to upload a revised version during the rebuttal period, but we will certainly improve our manuscript based on your invaluable suggestions. Also, we look forward to continuously improving our dataset and we will release an enhanced version 2.0 in the near future.
>
> If something is still unclear, welcome to continue the discussion with us. Once again, thank you very much for your insightful feedback!

---

> > ### Comment · Reviewer_f7Y3 · 2024-08-16
> > **Rating increased**
> >
> > I thank the authors for the detailed responses and insightful discussions! I have increased my rating to reflect the additional information and clarifications.

---

### Official Review · Reviewer_F2bD · 2024-07-25
**Valuable task, questions about benchmark construction**

**Rating:** 5
**Confidence:** 3
**Correctness:** Dataset construction is missing sever…

**Review:**

This work benchmarks recent language models in an interesting setting: statistical analysis of datasets. The reasoning behind this setting is well presented: much of benchmarking for mathematical abilities in current language models has focused on tasks where LMs have to both choose and execute to complete a task. Conversely, benchmarking for statistical analysis ability requires a model to have good domain knowledge and be able to infer the correct statistical test.

Overall, the manuscript includes an in-depth evaluation of model capabilities across a large range of statistical tests. Description of experimental results is comprehensive. The authors include a well though-out error analysis; perhaps the best part of the manuscript.

Constructing high quality evaluation data is expensive, and it is puzzling to see authors build this benchmark entirely through synthetic data generation. A better approach would have been to construct a small, high quality set of human curated task instance, and augment it via a synthetic data generation pipeline.

Besides the philosophy followed by this work, the methodology followed to construct the dataset could be better documented and evaluated. Specifically:
1. Steps 1-4 is challenging to understand without showing at least a few examples. How are what is used to synthesize the questions? is a OpenAI model? If so, what is the prompt? How did authors ensure that the model had adequate knowledge of the methods (after all, the whole point of this paper is to measure such knowledge).
2. How did authors check for errors at each steps? Did they perform manual validation? If so, what was the accuracy of each step? Did multiple authors check the same pipeline trace? if so, what was their agreement?
3. The question refinement in Step 5 is poorly described. Exactly, what kind of refinement was used here? Given GPT-3.5-Turbo is among the worst performing models in Table 2, why wasn't GPT-4o used?

Prompting strategies should have been discussed more extensively. For example, why only up to 1 shot is used? Why not 5? In detail, what is the domain knowledge approach? how was it derived?

I also would have expected larger open weight models to be included in the evaluation. Many of these models are cheaply available through a large number of API based vendors, just like OpenAI models are. Llama 3 70b, Qwen 72b, Mixtral 8x22B, or similar models should have been included.

Low performance of human experts in some categories might indicate issues in synthetic dataset construction. The manuscript says experts reviewed questions, but do not discuss how it was conducted.

**Strengths:**

- Important, well motivated task.
- Extensive result analysis section.
- Good error analysis.

**Additional Feedback:**

N/A

**Clarity:**

Dataset construction and experimental setup could be clearer. Rest is well presented.

**Documentation:**

Provided on Github.

**Ethics:**

Nothing to flag.

**Limitations:**

- Dataset contain no human-authored content
- Benchmark is fairly small at just a little over one thousand samples. Majority of the dataset is reserved for finetuning.

**Opportunities For Improvement:**

The dataset construction and experimental analysis sections could be significantly improved. In my review, I highlight several open questions.

**Relation To Prior Work:**

The related work section could be improved. The authors provide background on language models in general, where they should have focused more on related benchmark and LMs for scientific and mathematical applications.

**Summary And Contributions:**

This paper introduces StatQA, a benchmark for evaluating language models' ability to provide assistance with statistical analyses. StatQA is build using synthetic data generation. Several proprietary and open-weights models are evaluated on StatQA, using zero shot, few shot, and fine-tuning setups. The manuscript also include a comparison with human annotators, and a detailed analysis of results and failure causes of the benchmarked systems.

---

> ### Author Rebuttal · Authors · 2024-08-15
>
> Dear Reviewer,
>
> Thank you very much for your constructive suggestions. We are delighted that our motivation and analysis are recognized. Below, we will try our best to address raised issues.
>
> **1. About pipeline of dataset-building**
>
> We are pleased to explain the process of building our dataset, and we will revise based on your helpful feedback
>
> - Our synthesis pipeline is aligned with your insight to a great extent. In Section 2.3, we carefully prepared templates manually for expressions for each type of situation with examples in Table 4 in appendix Section B.3. We then fill templates with column headers to synthesize questions and refine them later. In this process, the manual preparation of the question templates is similar to curating a high-quality set of task instances, and the subsequent synthesis and refinement of originally synthesized questions can be considered as "augmentation".
> - An example for pipeline steps: start with Table=(C1, C2, ...), we set a target method set M, identifying columns that meet M’s prerequisites (Step 1-2). These columns fill into templates, e.g., “Is C1 independent of C2?” to generate original questions (Step 3). Then we balance and partition dataset, refine questions in Step 4-5.
>
> **2. How did the authors check?**
>
> Two authors are involved in the manual checking. As mentioned at the end of Section 2.3, the primary factors of our dataset quality are question templates and refinement, so our validation efforts were concentrated on question templates, refining results, and overall dataset quality.
>
> The criteria for agreement during manual checks included: 1) the representativeness and quality of question templates, and 2) ensuring that the refined questions maintained varied expressions without altering the original intent; 3) If intent changed, manually correct.
>
> **3. Step 5: question refinement**
>
> We use GPT-3.5-Turbo to refine and rephrase. The goals are to 1) correct potential English grammar mistakes (e.g. singular and plural forms); 2) make the sentence semantically coherent (e.g. mitigating the possible expression rigidity and artificiality when filling templates); 3) increase the diversity of expressions.
>
> - How to refine
>
> We prompt GPT-3.5-Turbo with goals above, and descriptive information (collected from source platform) of involved columns. Then, we ask it to rephrase the question sentence based on our requirements and description, without changing the original meaning. Please refer to the prompt in Table 5 in appendix Section C.3.
>
> - Why GPT-3.5-Turbo
>
> This task is more of a requirement for English proficiency and text processing instead of statistical knowledge. In practice, we examined the refined results, and we found although GPT-3.5-Turbo was not as strong as GPT-4 in statistical QA tasks, it still showed satisfactory English proficiency and has been very capable of handling relatively simple text processing tasks like sentence rephrasing here to meet goals above. To try to ensure both effectiveness and affordability, we chose GPT-3.5-Turbo.
>
>
> **4. More explanation for prompting strategies**
>
> We think with limited example, 0/1-shot can better reflect LLMs’ intrinsic and standard capabilities[1] on statistical QA. Besides, limited by llama-2 input token, more shots will overflow. To be fair, we uniformly use 1-shot.
>
> Domain knowledge in prompt refers to brief, hand-collated information about the usage scenarios and applicability of statistical methods, e.g. "F-Test for Variance: Whether there is a significant difference in variance between normally distributed variables", as shown in appendix Table 8.
>
> **5. Larger opensource LLMs**
>
> Thanks for the beneficial advice! Considering our funding and available APIs from vendors, we provide supplementary results with the following representative larger open-source LLMs. We found qwen2-72b performed strongly, comparable to the GPT models, which is exciting for users concerned about data privacy. Limited by length, we show overall 0-shot accuracy here and full table in attached PDF.
> |||
> | --- | --- |
> | llama3-70b | 27.09% |
> | qwen2-72b-instruct | 44.71% |
> | yi-34b-chat | 28.03% |
>
> **6. Human performance**
>
> As clarified in Part 2 of response, we checked the dataset to ensure questions in our dataset are properly expressed. In post-experiment interviews, we learned that participants (even graduate students in statistics) were also unfamiliar with the prerequisites of many methods. Together with the complexity of data, broad coverage of methods and questions, this made task challenging for them. We also discussed limitations of human experiments in appendix Section F, as  it is worth conducting more in-depth and extensive experiments in future.
>
> **7. Related work**
>
> Thank you for the suggestion! We present a detailed literature review in appendix Section A, reviewing more literature on scientific and math reasoning, and we will continue to improve this part.
>
> **8. Dataset scale**
>
> For a more efficient evaluation and to help users with limited resources, we provide mini-StatQA with 1163 examples, which we believe is sufficient to test LLMs’ capabilities if we refer to other renowned science or math reasoning benchmarks[1][2]. Also, StatQA with 11623 test examples is released for users with sufficient resources. For training, D_train (Section 2.3 Step 4) contains another 12672 examples from different source tables.
>
> [1] Mathematical Capabilities of ChatGPT
>
> [2] SciBench: Evaluating College-Level Scientific Problem-Solving Abilities of Large Language Models
>
> ---
>
> As per latest NeurIPS rules, we are unable to upload a revised version during the rebuttal period, but we will improve our manuscript based on your invaluable suggestions. Also, we look forward to continuously improving our dataset and we will release an enhanced version 2.0 in the near future.
>
> If something is still unclear, welcome to continue the discussion with us. Once again, thank you for your insightful feedback!

---

### Official Review · Reviewer_u58N · 2024-07-28
**Review of "Are Large Language Models Good Statisticians?"**

**Rating:** 9
**Confidence:** 4

**Review:**

Pros:
Originality & Significance: A qualitative and novel approach for curating the StatQA dataset which is critical for evaluating LLMs on statistical tasks & understanding. This opens up venues for potential work on improving LLMs as well as creating industrial tools for improving the productivity of professionals. Moreover, the dataset incorporate real-world data.
Quality & Clarity: The paper is well-written and well organized, making it easily comprehensible. The authors use very clear language, nice explanations with illustrative examples and convey their goals, methodologies, and findings effectively.
Human-AI Comparison: Human experiments allowed for a direct comparison between human and LLM performance, revealing interesting insights into their diverse strengths and weaknesses.

Cons:
Cost: Because of the inherent expense of curating datasets, the proposed approach has less human-supervision in curating the dataset. Neither did the authors try to use the most powerful LLMs in curating the questions for the dataset nor measured how well the LLM generated questions are closer to the template questions.
Limited Scope of StatQA: While StatQA is a great contribution, it focuses only on a limited set of statistical tasks and methods. Expanding the benchmark to include a wider range of tasks and methods would enhance its utility. But it can be targeted for the future.
Choice of Kaggle & Rdatasets: There is lack of explanation on why only Kaggle & Rdatasets are chosen and limited information on how the tables are sampled from each of these.

**Strengths:**

Novelty: The StatQA benchmark is essential and novel for measuring LLMs capabilities for statistical applications. The focus on applicability of the statistical methods is a crucial aspect which has been overlooked in many datasets across domains.

Comprehensive: The LLM evaluations performed in this paper are comprehensive i.e. utilizing widely popular techniques such as zero/few-shot, chain of thought, LoRA, In-context learning (i.e. domain knowledge) and set the benchmarks well.

Reliable yet cost-effective curation: The dataset curation pipeline is appreciative which ensures diversity & extensibility while being cost-effective and still supporting the statistical evaluation.

Experiments & Results: The findings in this are significant and have potential implications for future in statistical applications of LLMs. The identification of error patterns for both humans and LLMs, contrasting them, suggests potential avenues for improving LLM performance and developing more effective tools & strategies collaborating AI & humans both in reserach & industry.

**Additional Feedback:**

Commendable work, significant impact, very useful and unlocks more doors to statistical applications using LLMs

**Clarity:**

The authors pose a high degree of clarity in their submission. Overall, the key question, goal discussion, methodology, findings and conclusion are clearly communicated and organized intuitively. Moreover, the methodologies are meticulously detailed ensuring reproducibility. All the tables & figures summarise their findings very insightfully and clearly.

**Correctness:**

Almost all claims in the submission are correct. The dataset is constructed using a sound approach that ensures diversity, extensibility, qualitative while being cost-effective. Refer to the "opportunities for improvement" section for more details on where to improve.

**Documentation:**

The documentation is perfect.

**Ethics:**

Nothing significant.

**Limitations:**

Although there is reasonable discussion about limitations, a few missing things are:
1. Acknowledgement that the authors are aware of biases, fairnesses in the use of LLMs in both dataset curation and during LLM, human evaluation and should advise the community in similar manner.
2. [nit] Advise about utilizing the LLMs for practical use should consider the limitations of LLMs as they can generate potentially incorrect information.

**Opportunities For Improvement:**

Following are some opportunities for improvement overall in this paper:
1. The choice of GPT-3.5 turbo for rephrasing the templated questions is weak especially with GPT-4, Claude or other powerful LLMs around. It directly affects the quality of the dataset. Also there wasn't any measurement performed to understand how similar the rephrased questions are from the original. Scores such as BLEU, BERTScore, etc could be used to understand the translation quality and based on that the scores, the appropriate LLM can be chosen.
2. Human evaluation is another weak point in this paper and could be improved. The selected candidate pool is very narrow and hence the performance numbers can't be considered significant because of that. The candidate pool should be broader and evaluation with candidates with no statistical background maybe considered optional/redundant because of the cost.

I definitely understand that cost is a critical factor in curating these datasets but it is also essential to ensure quality for the datasets that are released to the public and of high significance.

**Relation To Prior Work:**

Yes.

**Summary And Contributions:**

The paper as titled "Are Large Language Models Good Statisticians?" proposes to understand the key question "Do LLMs understand statistics" or how well they do. New benchmarks have been released every day to measure/understand the capabilities of LLMs in different domains and for different objectives. Similarly, the contributions of this paper helps formalize the understanding around the capabilities of LLMs for diverse statistical analysis tasks. The key contributions of the paper include:
1. StatQA dataset, a first of its kind statistical analysis dataset focusing on the applicability assessment of statistical methods.
2. LLM evaluation & benchmark - Different LLMs are evaluated on the dataset using popular techniques such as 0-shot, 1-shot, chain-of-thought, domain knowledge & LoRA (parameter-efficient fine tuning) and a benchmark has been set.
3. Human evaluation & comparison - In addition to LLMs, humans are also utilized for evaluation to evaluate and compare the human performance accordingly.
4. Future direction/ Opportunity - The complementary nature of human performance and LLM performance on this dataset signifies the need for human-ai collaboration in both improving the statistical understanding of these LLMs as well as using them carefully in real-applications.

Overall, the authors adopted a reasonable & diverse yet cost-effective & extensible approach for curating the StatQA dataset and established benchmarks using both humans & AI for the same thereby opening a lot more opportunities for measuring & improving the statistical capabilities of LLMs.

---

> ### Author Rebuttal · Authors · 2024-08-15
>
> Dear Reviewer,
>
> Thank you very much for your time and constructive suggestions! We sincerely appreciate your recognition of our work, which is encouraging. Below, we will try our best to address the issues raised.
>
> **1. GPT-3.5-Turbo for rephrasing**
>
> - *Why GPT-3.5-Turbo as refiner*
>
>     Thank you for pointing out! In our manuscript, we use GPT-3.5-Turbo to refine and rephrase statistical questions without changing original meanings, the goals of this task are to 1) correct potential grammar mistakes (e.g. singular and plural forms); 2) make the sentence semantically coherent (e.g. mitigating the possible expression rigidity and artificiality when use templates); 3) increase the diversity of expressions. This task is more of a requirement for English proficiency and text processing instead of statistical knowledge. In practice, we examined the refined results, and we found although GPT-3.5-Turbo was not as strong as GPT-4 in statistical tasks, it still showed satisfactory English proficiency and has been very capable of handling relatively simple text processing tasks like sentence rephrasing here to meet our goals above.
>
> - *BLEU Score*
>
>     Following your helpful suggestions, we used the BLEU score on mini-StatQA to measure the similarity between the original and the refined questions: the lower quartile, median, and upper quartile BLEU scores are 0.037, 0.081, and 0.164, respectively, with a mean BLEU score of 0.121.
>
>     We are therefore inclined to think that the original questions from templates are adequately rephrased. We wholeheartedly agree with your emphasis on the importance of dataset quality for the community, and to try to ensure both effectiveness and affordability, we chose GPT-3.5-Turbo here. Also, we look forward to continuously improving our dataset and plan to release an enhanced version 2.0 in the near future.
>
>
> **2. Human experiments for non-statistical group**
>
> As discussed in the appendix (Section F, Limitation), we aware limitations posed by inability to recruit more participants. Our purpose in setting up non-statistical background group is trying to gain insights into how ordinary students approach our statistical problems. We hope this can inspire future research on human-LLM collaboration and contribute to the development of statistical analysis systems that are more accessible to non-specialists in future. Should additional funding become available, we would be eager to conduct more extensive human experiments.
>
> **3. Reason for choice of Kaggle & Rdatasets**
>
> Kaggle and Rdatasets are popular platforms that offer a wide variety of datasets from different domains. They are extensively used in educational projects and data competitions related to mathematical modeling, statistical analysis, and beyond. We carefully selected 78 popular tables from these platforms, frequently used in practice and competitions, covering diverse domains and varying in size (including both large and small samples). We also ensured that most tables had a sufficient number of columns (>5) for subsequent extraction.
>
> **4. Additional aspects for the limitation section**
>
> Many thanks for your insightful comments. We also agree that the LLMs may introduce issues of bias and fairness, as well as the possibility that LLMs may generate potentially erroneous or even toxic information. Therefore, we will add these discussions to the limitation section, and thank you for pointing them out!
>
> ---
>
> As per latest NeurIPS rules, we are unable to upload a revised version during the rebuttal period, but we will certainly improve our manuscript based on your invaluable suggestions, and we will release a enhanced version 2.0 dataset soon.
>
> Once again, thanks a lot for your time and insightful feedback!

---

> > ### Comment · Reviewer_u58N · 2024-08-16
> > **Response to Authors**
> >
> > Thanks for the detailed information. Adding the aforementioned points to the manuscript would not change my original rating because my original rating is still based on the overall quality of the paper without these non-major concerns.

---

### Author Rebuttal · Authors · 2024-08-15

# Global Response
We sincerely thank all the reviewers for their efforts and insightful suggestions. We are delighted and encouraged that our work is found to be important, gap-filling, interesting, and well-written, and that our motivation, analysis, and comparative study are recognized.

According to the latest NeurIPS rules, although we are unable to upload a revised version at the rebuttal/discussion stage this year, we will revise our manuscript based on these invaluable suggestions from reviewers. To clarify and address some concerns, we will revise our manuscript according to the feedback, which includes but not limited to:

1. **More explanation about choosing GPT-3.5-Turbo as the refiner.** Some reviewers are curious as to given GPT-3.5-Turbo did not perform as well as GPT-4 in our statistical QA tasks, why we use it to refine and rephrase question expressions. Actually, these are two different tasks. For question refining, our goals are to 1) correct potential grammar mistakes; 2) improve semantic coherency; and 3) increase the diversity of expressions. This task is more of a requirement for English proficiency and text processing instead of statistical knowledge. In practice, we examined and found GPT-3.5-Turbo has been very capable of handling relatively simple text processing tasks here to meet our goals above, and it has better affordability. We will add this part of elaboration to manuscript.
2. **Clarify the scale of dataset.** We noticed that there are some small misunderstandings on the dataset scale. In fact, we use mini-StatQA (1163 examples) for more efficient evaluation, which is stratified sampling from StatQA, ensuring they are distributionally proximate. We believe mini-StatQA has sufficient test examples to evaluate LLMs’ capabilities if referring to other renowned science or math reasoning benchmarks. Also, the full StatQA with 11623 examples is also released and available to facilitate those with very sufficient computational resources. We will further clarify this point in our manuscript.
3. **Improve the elaboration of the steps in dataset construction to make it clearer.** As advised by reviewer F2bD, jPG1, f7Y3, we think it would be beneficial to introduce an example to help understand and provide more information on dataset construction pipeline, thanks for their feedback!
4. **Additional experiments.** We thank reviewers u58N, F2bD, and 8PyK for their helpful suggestions and add the following experiments:
    - Following reviewer u58N’s useful advice, we used the BLEU Score to evaluate and found original questions have been adequately rephrased;
    - Evaluation of larger open-source LLMs. As advised by reviewer F2bD, we have evaluated representative llama3-70b, qwen2-72b-instruct, and yi-34b-chat, and qwen2-72b-instruct showed strong performance, which is exciting for potential users concerned about data privacy;
    - Evaluation of smaller open-source LLMs. Thanks to reviewer 8PyK for his/her attention on users only having access to very limited resources. We have also tested on much smaller open-source LLMs: qwen2-0.5b-instruct and qwen2-1.5b-instruct. While qwen2-0.5b-instruct is incompetent, 1.5b-instruct is comparable to some medium LLaMA models, which is encouraging for those with limited resources.

    We will include these additional results and discussion in our manuscript.

5. **Other improvements.** We appreciate all reviewers for their comments and suggestions on other aspects, like literature review, limitations, prompting strategies, pointing out typos, etc.
**We have uploaded separate responses for each reviewer to include more details and elaborations, please refer to them.**

We believe that our manuscript will benefit from all these constructive suggestions. Also, we look forward to continuously improving our dataset and we will release an enhanced version 2.0 in the near future. Once again, many thanks to all the reviewers!

---

### Decision · Program_Chairs · 2024-09-26

**Decision:**

Accept (Poster)

**Comment:**

The reviewers seem largely in agreement that this is a good paper (scores ranging from 5.0-9.0), that warrants acceptance at the conference. Reviewer F2bD raises some fair questions, which have been addressed by the author. It is unclear whether the reviewer is satisfied with the reply, but given the largely positive response by other reviewers, I will recommend acceptance.